# Administration of anti-HIV-1 broadly neutralizing monoclonal antibodies with increased affinity to Fcγ receptors during acute SHIV$_{AD8-EO}$ infection

Anti-HIV-1 broadly neutralizing antibodies (bNAbs) have the dual potential of mediating virus neutralization and antiviral effector functions through their Fab and Fc domains, respectively. So far, bNAbs with enhanced Fc effector functions in vitro have only been tested in NHPs during chronic simian-HIV (SHIV) infection. Here, we investigate the effects of administering in acute SHIV$_{AD8-EO}$ infection either wild-type (WT) bNAbs or bNAbs carrying the S239D/I332E/A330L (DEL) mutation, which increases binding to FcγRs. Emergence of virus in plasma and lymph nodes (LNs) was delayed by bNAb treatment and occurred earlier in monkeys given DEL bNAbs than in those given WT bNAbs, consistent with faster clearance of DEL bNAbs from plasma. DEL bNAb-treated monkeys had higher levels of circulating virus-specific IFNγ single-producing CD8+ CD69+ T cells than the other groups. In LNs, WT bNAbs were evenly distributed between follicular and extrafollicular areas, but DEL bNAbs predominated in the latter. At week 8 post-challenge, LN monocytes and NK cells from DEL bNAb-treated monkeys upregulated proinflammatory signaling pathways and LN T cells downregulated TNF signaling via NF-κB. Overall, bNAbs with increased affinity to FcγRs shape innate and adaptive cellular immunity, which may be important to consider in future strategies of passive bNAb therapy.

To date, no effective and widely available approach to cure human immunodeficiency virus type 1 (HIV-1) infection has been found. Daily antiretroviral therapy (ART) is the current standard of care that blocks new rounds of viral replication and can reduce plasma viral load to undetectable levels while effectively limiting or eliminating potential for transmission[1,2]. However, HIV-1 persists despite ART in infected cell populations, which contribute to viral reservoirs[3–5] and lead to rapid rebound of plasma viremia once treatment is discontinued[6,7]. Thus, it is imperative to find new therapeutic interventions for HIV-1 infection.

Over the past decade, potent anti-HIV-1 broadly neutralizing antibodies (bNAbs) have been identified and gained considerable attention as tools for HIV-1 prevention and treatment due to their ability to neutralize a wide variety of HIV-1 strains with high potency. Anti-HIV-1 bNAbs target different sites of the HIV-1 envelope (Env) glycoprotein such as the CD4-binding site[8–11], the variable regions 1 and 2 (V1/V2)[12–14], the V3-glycan site[15–17], and the membrane proximal external region[18]. Alone or in combination, they have been tested in strategies for treatment of simian-HIV (SHIV) infection in non-human primates (NHPs) and HIV-1 infection in humanized mice and humans.

✉ e-mail: rkoup@mail.nih.gov

bNAbs administered to NHPs or humans in the chronic phase of infection were well tolerated and had a robust, but transient, antiviral activity, as the decline in plasma viral load observed shortly after their administration was followed by viral rebound to pre-treatment levels[19–25]. bNAb treatment initiated in the acute phase of infection in infant and adult NHPs led to efficient reduction and, in some cases even suppression, of viral replication[26–29]. In one of those studies, Nishimura et al.[29] treated SHIV_{AD8-EO}-infected NHPs at days 3, 10, and 17 post-challenge with a combination of the wild-type (WT) bNAbs 3BNC117 and 10–1074, which target the CD4-binding site and V3-glycan site on the HIV-1 Env, respectively[10,15]. This treatment regimen resulted in long-term suppressed plasma viremia in about half of the treated animals in a CD8$^+$ T cell-dependent manner, as evidenced by rapid reemergence of viremia in controller animals following CD8$^+$ T cell depletion[29]. These findings led to the hypothesis that the presence of bNAbs early in infection and before natural antibodies occur may lead to the formation of immune complexes, which, following cross-presentation to CD8$^+$ T cells, may boost antiviral immune responses[30] and lead to viral control.

The Fab domain of antiviral monoclonal antibodies (mAbs) can mediate the neutralization of a virus, thereby blocking its entry into cells. In addition, the Fc domain can recognize and clear virus-infected cells by engaging host immune components in Fc-mediated effector functions, such as antibody-dependent cellular cytotoxicity (ADCC), phagocytosis (ADCP), and complement-dependent cytotoxicity[31]. Several studies using Fc domain-engineered anti-HIV-1 bNAbs suggested that Fc-mediated effector functions contribute to the in vivo antiviral activity of bNAbs in humanized mice[32–35]. Despite conflicting results obtained from studies in NHPs[36–39], two recent studies quantified the relative contribution of Fc-mediated *vs.* Fab-mediated functions to the antiviral activity of anti-HIV-1 bNAbs and reported that the former accounted for about 20 to 45% of the overall activity[40,41]. In these studies, anti-HIV-1 bNAbs with enhanced or impaired Fc-mediated effector functions in vitro were infused in HIV-1-infected humanized mice and chronically SHIV-infected NHPs[40,41]. However, no study thus far has investigated the effects of administering anti-HIV-1 bNAbs with enhanced Fc effector functions in the acute phase of SHIV or HIV infection.

In this study, we evaluated the virological and immunological effects of administering early after SHIV_{AD8-EO} infection either WT bNAbs or bNAbs carrying the S239D/I332E/A330L (DEL) mutation, a mutation introduced in the Fc domain that increases binding affinity to rhesus FcγRIII and FcγRII and enhances Fc-mediated effector functions in vitro[40,42]. Notably, we extended our investigation to the lymph nodes (LNs), where we studied in detail the presence and localization of infused bNAbs as well as their effects at the cellular level. WT or DEL bNAbs were administered at days 3, 10, and 17 post-intrarectal SHIV_{AD8-EO} challenge, similar to the early WT bNAb therapy regimen followed by Nishimura et al.[29] Treatment of monkeys with DEL bNAbs allowed us to additionally test if early administration of bNAbs with increased affinity to FcγRs could boost antiviral effects. We used a combination of VRC07-523-LS and PGT121, which target non-overlapping epitopes on the HIV-1 Env[17,43] and have similar pharmacokinetic (PK) profiles upon intravenous infusion into NHPs[20,21,43,44]. VRC07-523 is an engineered, more potent variant of VRC01 that targets the CD4-binding site of the HIV-1 Env[9,43], and the LS (M428L/N434S) mutation introduced in its Fc domain extends its plasma half-life in NHPs[43]. PGT121 is a potent bNAb that targets glycan-dependent V3 epitopes of the HIV-1 Env[17]. In monkeys infected with SHIV_{AD8-EO} and subjected to this early bNAb therapy regimen, emergence of virus in plasma and LNs was delayed proportionally to the duration of the infused bNAbs in circulation. DEL bNAb-treated monkeys developed distinct circulating virus-specific CD8$^+$ T cell responses and LN immune cell signatures that set them apart from monkeys that received WT bNAbs or were left untreated.

## Results

### Early passive bNAb therapy suppressed acute-stage plasma viremia in SHIV_{AD8-EO}-infected rhesus macaques

To evaluate the therapeutic effect of administering a combination of anti-HIV-1 bNAbs during acute infection, 30 Indian origin rhesus macaques were inoculated intrarectally with 1000 median tissue culture infective doses (TCID$_{50}$) of SHIV_{AD8-EO}. Monkeys were then either left untreated (control group, $n = 10$) or intravenously infused at days 3, 10, and 17 post-challenge with 10 mg/kg each of VRC07-523-LS and PGT121 (WT bNAb treatment, $n = 10$) or VRC07-523-LS/DEL and PGT121/DEL (DEL bNAb treatment, $n = 10$) (Fig. 1A). VRC07-523-LS and PGT121 neutralization of the SHIV_{AD8-EO} challenge stock was similar to that of 3BNC117 and 10-1074, respectively, which were previously used by Nishimura et al. in a similar therapeutic strategy[29] (Table 1). In addition, the DEL mutation did not affect neutralization of SHIV_{AD8-EO} (Table 1).

Peripheral blood samples were collected from these animals at multiple timepoints throughout 17 months to monitor plasma viremia. Viremia was detected in untreated monkeys within, on average, the first 2 weeks post-challenge (Fig. 1, B and C left). In contrast, in treated monkeys, viremia was overall suppressed for the first 10 weeks post-challenge upon WT bNAb treatment and 6 weeks post-challenge upon DEL bNAb treatment (Fig. 1, B, C left). Thus, both the time to first measurable plasma virus leading to persistent infection and the time to reach peak viremia were significantly longer in the WT bNAb-treated group than in the untreated group ($p < 0.001$ for both comparisons) (Fig. 1C). However, the peak and set-point viral loads (Fig. 1D) and the overall plasma viremia as measured by area under the curve (AUC) (Fig. 1E) did not vary significantly with any of the bNAb treatments. Taken together, these data show that triple administration of bNAbs targeting two distinct and non-overlapping viral epitopes in the acute phase of SHIV_{AD8-EO} infection delays onset of plasma viremia without affecting the overall plasma viral burden.

One untreated monkey, two WT bNAb-treated monkeys, and three DEL bNAb-treated monkeys did not develop plasma viremia for more than 30 weeks post-challenge, as evaluated by standard (Fig. 1, B and D right) and ultrasensitive (Supplementary Table 1) SIV Gag RNA quantitative RT-PCR (qRT-PCR) with detection limits of 15 and 1 copies/mL, respectively. To ascertain the infection status of these monkeys, we depleted their CD8$^+$ T cells through intravenous infusion of the anti-CD8β mAb CD8b255R1, which specifically targets macaque CD8$^+$ T cells (Supplementary Fig. 1A). Circulating CD8$^+$ T cells were selectively depleted during the 36 weeks that followed anti-CD8β mAb infusion (Supplementary Fig. 1, B, C). In contrast, the levels of other T cell subsets, B cells, monocytes, and NK cells quickly returned to baseline after the first 6 h of anti-CD8β mAb infusion (Supplementary Fig. 1, B–D). Despite efficient depletion of CD8$^+$ T cells, no burst in plasma viremia occurred up to 21 days post-mAb infusion (Supplementary Table 2). We also did not detect cell-associated SIV Gag DNA or RNA in peripheral blood mononuclear cells (PBMCs) from the DEL bNAb-treated, anti-CD8β mAb-infused monkeys up to 21 days post-mAb infusion (Supplementary Table 3). Therefore, we concluded that these monkeys never became infected and that the SHIV_{AD8-EO} stock used in this study, when given intrarectally at the dose of 1000 TCID$_{50}$, does not infect all monkeys. Unless otherwise indicated, these uninfected monkeys were excluded from further analyses.

Of note, this study was performed in two separate batches, the first with 6 monkeys per group (depicted in full gray lines in Fig. 1B and circles in Fig. 1, C–E) and the second with 4 monkeys per group (depicted in dashed lines in Fig. 1B and squares in Fig. 1, C–E). Plasma viremia did not segregate the animals into the two different batches (Fig. 1, B–E), thus viremic monkeys from both batches were grouped together for all subsequent analyses.

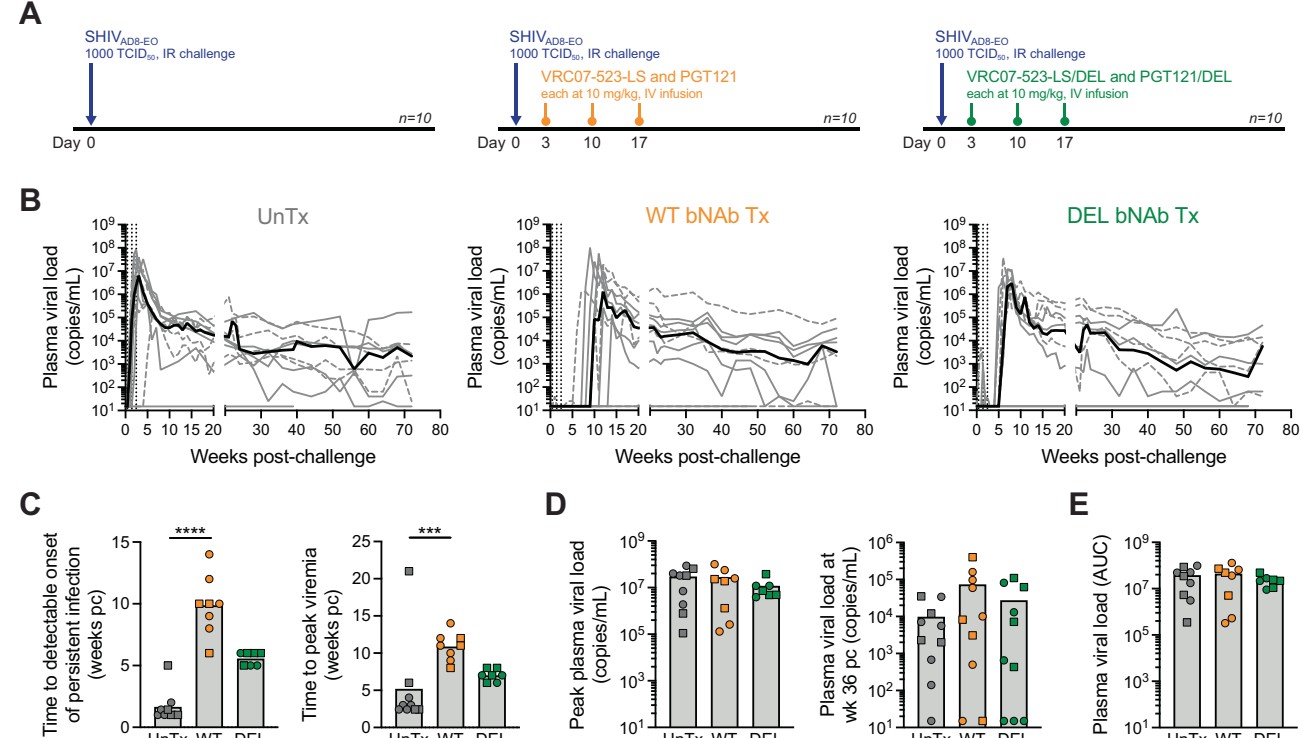

**Fig. 1 | Study design and therapeutic efficacy of anti-HIV-1 bNAb combinations.**
**A** Thirty rhesus macaques were inoculated intrarectally with SHIV_{AD8-EO} and either left untreated (n = 10) (left) or treated at days 3, 10, and 17 post-challenge with a combination of either VRC07-523-LS and PGT121 (WT bNAbs, n = 10) (middle) or VRC07-523-LS/DEL and PGT121/DEL (DEL bNAbs, n = 10) (right). Each bNAb was infused intravenously at 10 mg/kg. **B** Plasma viral load up to 72 weeks post-challenge in untreated (left), WT bNAb-treated (middle), and DEL bNAb-treated (right) monkeys. Gray and black lines represent individual and median plasma viral loads, respectively. Vertical dotted lines indicate the timings of bNAb infusions. **C** Time to first detectable plasma virus leading to persistent infection (left) and to peak plasma viremia (right) in untreated and bNAb-treated monkeys. N = 9, 8, and 7 untreated, WT bNAb-treated, and DEL bNAb-treated monkeys, respectively. P < 0.0001 (left) and p = 0.0007 (right). **D** Peak plasma viral load (left) and plasma viral load at week 36 post-challenge (right) in untreated and bNAb-treated

monkeys. N = 9, 8, and 7 (left) and n = 10, 10, and 10 (right) untreated, WT bNAb-treated, and DEL bNAb-treated monkeys, respectively. **E** Plasma viral load throughout the first 72 weeks post-challenge in untreated and bNAb-treated monkeys as determined by AUC analysis. N = 9, 8, and 7 untreated, WT bNAb-treated, and DEL bNAb-treated monkeys, respectively. Individual data from the initial 6 monkeys per group are indicated with gray solid lines (**B**) and circles (**C–E**), whereas data pertaining to the later 4 monkeys per group are denoted with gray dashed lines (**B**) and squares (**C–E**) (see text for more details on the two monkey batches). Bar graphs show the mean and individual datapoints (**C–E**). The Kruskal–Wallis test followed by Dunn's multiple comparison test was used to detect significant differences between monkey groups. N indicates the number of biological replicates (monkeys). Source data are provided in the Source Data file. AUC area under the curve, IR intrarectal, IV intravenous; pc post-challenge, TCID_{50} median tissue culture infective doses, Tx treatment, UnTx untreated, wk week.

## bNAbs persisted in plasma throughout the period of infusions and progressively declined thereafter as strong anti-drug antibody (ADA) responses developed

Given the different patterns of virus emergence between WT and DEL bNAb-treated monkeys, we hypothesized that the circulating titers of WT and DEL bNAbs might differ. To test this hypothesis, we measured longitudinally by enzyme-linked immunosorbent assay (ELISA) the plasma concentration of the infused bNAbs in all animals (both infected and uninfected) up to 14 weeks post-challenge. Right after the first infusion, all bNAbs reached plasma levels that well exceeded their IC_{80} values for in vitro neutralization against SHIV_{AD8-EO} (Fig. 2A and Supplementary Fig. 2A). bNAb levels increased after each infusion timepoint, with VRC07-523-LS and PGT121 reaching their average peak concentrations of 113 and 139 μg/mL, respectively, at week 3 post-challenge (i.e., 4 days after the third infusion), and VRC07-523-LS/DEL and PGT121/DEL reaching their average peak concentrations of 15 and 56 μg/mL, respectively, at week 2 post-challenge (i.e., 4 days after the second infusion). Overall levels of the WT bNAbs as measured by AUC were significantly higher than of their corresponding DEL mutants (p < 0.001 for both VRC07-523-LS vs. VRC07-523-LS/DEL and PGT121 vs. PGT121/DEL) (Fig. 2B). In addition, while the WT bNAbs had similar PK profiles, the levels of PGT121/DEL were significantly higher than those of VRC07-523-LS/DEL (p < 0.001) (Fig. 2B). Shortly after the third

infusion, bNAb levels started declining over time (Fig. 2A and Supplementary Fig. 2A). VRC07-523-LS and PGT121 reached undetectable levels within 5–12 and 6–11 weeks post-challenge, respectively, with the exception of one monkey in which both bNAbs were still detectable 14 weeks after challenge (Supplementary Fig. 2A). VRC07-523-LS/DEL and PGT121/DEL remained in circulation for a shorter time, reaching undetectable levels within 2–5 and 4–9 weeks post-challenge, respectively (Supplementary Fig. 2A).

Clearance of bNAbs from the plasma occurred concomitantly with the development of strong ADA responses (Fig. 2C and Supplementary Fig. 2B). ADA responses against the DEL bNAbs were significantly stronger than those against their WT counterparts (p = 0.009 for VRC07-523-LS vs. VRC07-523-LS/DEL and p = 0.003 for PGT121 vs. PGT121/DEL) (Fig. 2D), consistent with the faster clearance of DEL bNAbs from circulation. As expected, the overall magnitude of ADA responses inversely correlated with the duration of bNAbs in plasma (Spearman's r = −0.8273 and p < 0.0001 for VRC07-523-LS and VRC07-523-LS/DEL, and Spearman's r = −0.9195 and p < 0.0001 for PGT121 and PGT121/DEL) (Supplementary Fig. 2C), which in turn strongly correlated with the time to onset of persistent infection (Spearman's r = 0.8991 and p < 0.0001 for VRC07-523-LS and VRC07-523-LS/DEL, and Spearman's r = 0.9158 and p < 0.0001 for PGT121 and PGT121/DEL) (Supplementary Fig. 2D).

To determine whether the virus emerged due to the decline in bNAb levels or to the development of mutations conferring resistance to the infused bNAbs, we sequenced and analyzed the diversity of SHIV$_{AD8-EO}$ Env at the time of peak viremia. An average of 1160 full-length single-genome Env sequences per animal (range 286–2329) were recovered via high-throughput single-genome sequencing (HT-SGS)[46]. About half of the animals available for this analysis retained the

WT virus sequence at peak viremia (Supplementary Fig. 2E) and its frequency did not vary significantly across the monkey groups (Supplementary Fig. 2F). Haplotypes that, in some monkeys, completely overtook the WT sequence included mutations located outside the CD4-binding sites and the V3 loop of the virus Env (Supplementary Fig. 2E and Supplementary Table 4) and were unlikely to represent escape mutations to the infused bNAbs.

Taken together, these data show that bNAbs infused early after SHIV$_{AD8-EO}$ infection persisted in circulation throughout the time of infusions without inducing selective pressure on the virus, which only emerged as strong ADA responses developed and bNAb levels went down.

### Table 1 | Neutralization potency of anti-HIV-1 bNAbs against the SHIV$_{AD8-EO}$ challenge stock

| bNAb | IC$_{50}$ (µg/mL) | IC$_{80}$ (µg/mL) |
| --- | --- | --- |
| VRC07-523-LS[a] | 0.723 | 1.570 |
| VRC07-523-LS/DEL[a] | 0.731 | 1.680 |
| PGT121[a] | 0.167 | 0.375 |
| PGT121/DEL[a] | 0.168 | 0.413 |
| 3BNC117[b] | 0.517 | 1.000 |
| 10-1074[b] | 0.210 | 0.389 |

[a]Dilution points for each bNAb were run in duplicate. bNAbs were tested against SHIV$_{AD8-EO}$ in at least 5 experiments, with no more than 2-fold variability between them; representative IC$_{50}$ and IC$_{80}$ values are shown.
[b]Data for 3BNC117 and 10–1074 were obtained from Pegu et al., Cell Host & Microbe, 2019.

### All infused bNAbs potently neutralized the challenge virus SHIV$_{AD8-EO}$ ex vivo

To ensure that the infused bNAbs in circulation retained their capacity to neutralize SHIV$_{AD8-EO}$, we measured ex vivo the neutralizing activity of plasma from both infected and uninfected monkeys against the challenge virus SHIV$_{AD8-EO}$. We used virus SIVmac251.30.SG3 as negative control and viruses 00836-2.5 and X2088.c9, each of them sensitive to only one of the two infused bNAbs (Supplementary Table 5), to individually assess the neutralizing capacity of each bNAb. Plasma samples from treated animals showed high neutralizing titers (ID$_{50}$ and ID$_{80}$ values) against

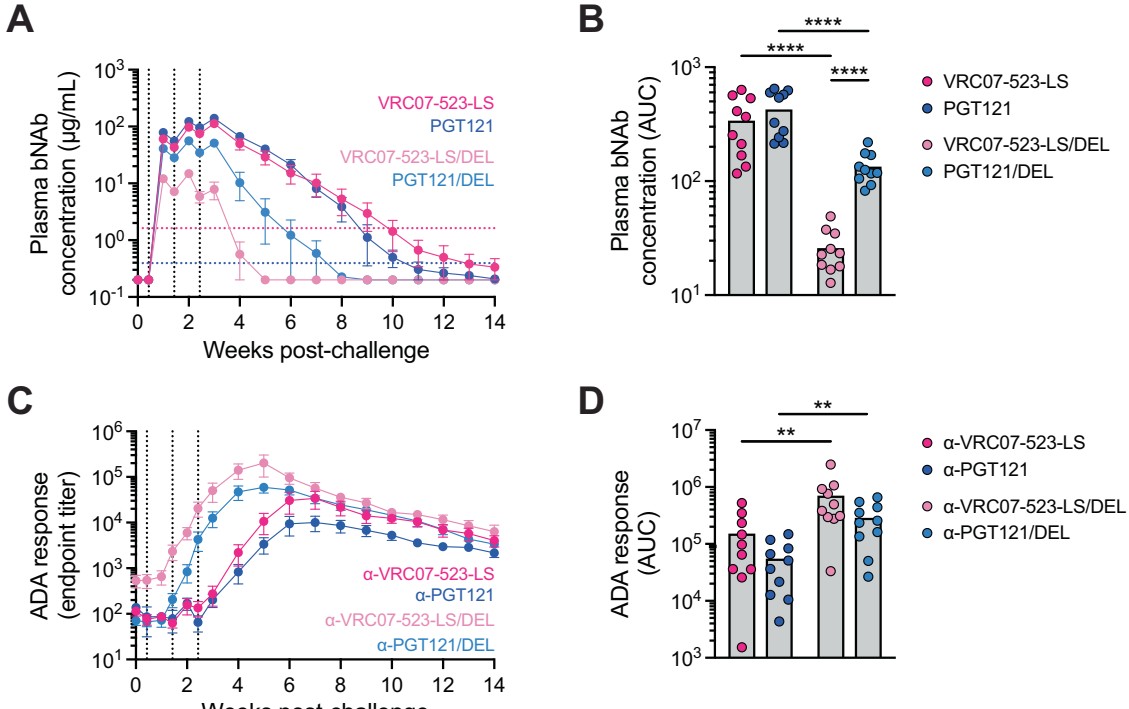

**Fig. 2 | PK of infused bNAbs and ADA responses. A** Concentration of VRC07-523-LS (dark pink), PGT121 (dark blue), VRC07-523-LS/DEL (light pink), and PGT121/DEL (light blue), as measured by ELISA in the plasma of monkeys that received either the WT or DEL bNAbs at days 3, 10, and 17 post-SHIV$_{AD8-EO}$ challenge. Vertical dotted lines indicate the timings of bNAb infusions and horizontal dotted lines indicate the in vitro neutralization IC$_{80}$ values against SHIV$_{AD8-EO}$ averaged for VRC07-523-LS and VRC07-523-LS/DEL (pink), and PGT121 and PGT121/DEL (blue). N = 10 for each bNAb and each timepoint except for VRC07-523-LS and PGT121 at week 0 and VRC07-523-LS/DEL and PGT121/DEL at weeks 10–14: n = 4; PGT121 at week 9: n = 9. **B** Plasma concentration of each infused bNAb up to 14 weeks post-challenge as determined by AUC analysis. N = 10 for each bNAb. Each P < 0.0001, two-sided p-values. **C** ADA response against VRC07-523-LS (dark pink), PGT121 (dark blue), VRC07-523-LS/DEL (light pink), and PGT121/DEL (light blue), as measured by ELISA in the plasma of monkeys that received either the WT or DEL bNAbs at days 3, 10,

and 17 post-SHIV$_{AD8-EO}$ challenge. Vertical dotted lines indicate the timings of bNAb infusions. N = 10 for each bNAb and each timepoint except for α-VRC07-523-LS at weeks 0 and 1, α-PGT121 at week 0, and α-VRC07-523-LS/DEL at week 6: n = 9; α-VRC07-523-LS/DEL at week 4: n = 8; α-VRC07-523-LS at days 10 and 17 and α-PGT121 at days 3, 10, and 17: n = 4, and α-VRC07-523-LS at day 3: n = 3. **D** ADA response against each infused bNAb up to 14 weeks post-challenge as determined by AUC analysis. N = 10 for each bNAb. P = 0.0089 for α-VRC07-523-LS vs. α-VRC07-523-LS/DEL and p = 0.0029 for α-PGT121 vs. α-PGT121/DEL, two-sided p-values. Graphs show the mean±SEM (**A**, **C**) and the mean and individual datapoints (**B**, **D**). The Mann-Whitney test was used to detect significant differences between the WT bNAbs, the DEL bNAbs, or the WT and DEL mutant of the same bNAb. N indicates the number of biological replicates (monkeys). Source data are provided in the Source Data file. ADA anti-drug antibody; AUC area under the curve.

SHIV$_{AD8-EO}$, 00836-2.5, and X2088.c9 throughout the first 3 weeks post-challenge (Supplementary Fig. 3, A and B), consistent with the PK profile of these bNAbs (Fig. 2A and Supplementary Fig. 2A). Interestingly, despite the lower plasma concentrations of DEL bNAbs (Fig. 2B), plasma from these monkeys neutralized both SHIV$_{AD8-EO}$ and X2088.c9 to a similar extent as plasma from WT bNAb-treated monkeys, and only 00836-2.5 to a significantly lower extent ($p = 0.03$ for both ID$_{50}$ and ID$_{80}$ AUC) (Supplementary Fig. 3C). The predicted plasma neutralizing titers, derived from the measured plasma bNAb concentrations and the in vitro neutralizing activity of each bNAb, very closely matched the experimental neutralizing titers (Supplementary Fig. 3A, B). Together, these data showed that each of the infused bNAbs retained their capacity to neutralize the challenge virus despite the existence of ADA responses.

We also tested the plasma samples from infected monkeys for endogenous neutralization breadth against a wide panel of HIV-1 Env pseudoviruses from clades A, B, and C (Supplementary Tables 6 and 7). Plasma samples were tested at weeks 64 and 116 post-challenge, long after the infused bNAbs had been cleared from the circulation. Both untreated and treated monkeys developed heterologous clade B neutralization as well as some levels of cross-clade neutralization (Supplementary Tables 6 and 7). However, the neutralization scores, calculated to summarize the plasma neutralizing capacity against all tested viruses, were not higher upon bNAb treatment (Supplementary Tables 6 and 7 and Supplementary Fig. 3D). Thus, early bNAb administration did not improve long-term neutralizing responses in treated monkeys.

### Early bNAb treatment delayed the emergence of SHIV$_{AD8-EO}$ RNA in LNs

We next investigated the effect of early bNAb treatment in LN viremia. LN sections were stained using the RNAscope in situ hybridization technology (Fig. 3A), and the levels of SHIV$_{AD8-EO}$ RNA$^+$ cells and SHIV$_{AD8-EO}$ RNA$^+$ virions were quantified and normalized to the LN areas for accurate comparisons of monkey groups. The total LN areas analyzed did not differ considerably among groups (Supplementary Fig. 4A). At week 2 post-challenge, SHIV$_{AD8-EO}$ RNA$^+$ cells and virions were detected in the LNs of the two untreated monkeys available for this analysis, whereas their levels were either null or very low in monkeys that received either WT or DEL bNAbs (Fig. 3B). In contrast, at week 8 post-challenge, SHIV$_{AD8-EO}$ RNA$^+$ cells and virions were detected in DEL bNAb-treated monkeys in addition to untreated monkeys but were still either undetectable or very low in monkeys that received the WT bNAbs ($p = 0.04$ for WT vs. DEL bNAb-treated groups for both SHIV$_{AD8-EO}$ RNA$^+$ cells and virions) (Fig. 3B). This virus detection pattern in LNs generally matched the emergence of virus in plasma, which occurred at approximately weeks 2, 6, and 10 post-challenge in untreated, DEL bNAb-treated, and WT bNAb-treated monkeys, respectively (Fig. 1B, C left). In fact, plasma viral loads strongly correlated with the levels of LN SHIV$_{AD8-EO}$ RNA$^+$ cells (Spearman's r = 0.8760, $p < 0.0001$) and virions (Spearman's r = 0.7632, $p < 0.0001$) at weeks 2 and 8 post-challenge (Supplementary Fig. 4B, C).

To determine the localization of SHIV$_{AD8-EO}$ RNA in the LNs of untreated and DEL bNAb-treated monkeys, SHIV$_{AD8-EO}$ RNA$^+$ cells and virions were quantified in the follicular and extrafollicular areas, which were defined by the presence and absence of CD20 expression, respectively (Fig. 3A). At week 2 post-challenge, viral events were similarly distributed across the LN follicular and extrafollicular areas in untreated monkeys (average percentage of follicular SHIV$_{AD8-EO}$ RNA$^+$ cells and virions: 56.0% and 52.4%, respectively), whereas at week 8 post-challenge, they tended to predominantly concentrate in the follicles in both untreated and DEL bNAb-treated monkeys (average percentage of follicular events: 75.5% and 65.6% for SHIV$_{AD8-EO}$ RNA$^+$ cells and 65.2% and 74.5% for SHIV$_{AD8-EO}$ RNA$^+$ virions in untreated and DEL bNAb-treated monkeys, respectively) (Fig. 3C, D). Similar to the

total LN area, the follicular and extrafollicular areas that were analyzed did not significantly differ across monkey groups (Supplementary Fig. 4D).

Altogether, these data indicate that, like in plasma, early bNAb treatment delayed the emergence of virus in LNs, and that the location of SHIV$_{AD8-EO}$ RNA$^+$ cells and virions in the LNs may vary during the course of the infection.

### Infused WT and DEL bNAbs were differently located in the LNs

Next, we investigated if infused bNAbs were also present in the LNs of treated monkeys (both infected and uninfected). LN sections from weeks 2 and 8 post-challenge (i.e., 4 days after the second bNAb infusion and approximately 5 weeks after all 3 infusions, respectively) were stained for confocal microscopy (Fig. 4A). Antibodies that recognize either the kappa or lambda light chain of human antibodies were used to identify VRC07-523-LS and VRC07-523-LS/DEL, which express kappa light chains[9,43], and PGT121 and PGT121/DEL, which express lambda light chains[47] (Fig. 4A). Importantly, these antibodies did not cross-react to rhesus antibodies. Like in the RNAscope analysis, the numbers of bNAb$^+$ cells were normalized to the LN areas, which did not vary considerably across monkey groups (Supplementary Fig. 4E). WT and DEL bNAbs were detected in LNs at weeks 2 and 8 post-challenge (Fig. 4B). In some monkeys, the levels of DEL bNAb$^+$ cells were higher than those of WT bNAb$^+$ cells, consistent with the higher affinity of DEL bNAbs to some rhesus FcγRs[40], and that difference reached statistical significance at week 8 post-challenge ($p = 0.009$) (Fig. 4B). Interestingly, bNAb$^+$ cell levels were comparable between weeks 2 and 8 post-challenge in each bNAb-treated group (Fig. 4B), in contrast to plasma bNAb levels, which were much lower at week 8 than week 2 post-challenge (Fig. 2A). In a separate analysis where we segregated cells bound with VRC07-523-LS from those bound with PGT121 (and the same for the DEL mutants), we found that the two bNAbs infused into monkeys were always bound to similar numbers of cells in the LNs (Fig. 4C).

We were also interested in understanding where bNAbs were located in the LNs; thus, we quantified bNAb$^+$ cells in the LN follicular and extrafollicular areas. While DEL bNAbs bound to more cells in the LNs (Fig. 4A), they tended to localize more in the extrafollicular areas than the WT bNAbs, which were more evenly distributed between the follicular and extrafollicular areas at any of the evaluated timepoints (average percentage of follicular VRC07-523-LS$^+$ cells and PGT121$^+$ cells: 44.5% and 44.6% at week 2 post-challenge, respectively, and 40.9% and 47.3% at week 8 post-challenge, respectively; average percentage of follicular VRC07-523-LS/DEL$^+$ cells and PGT121/DEL$^+$ cells: 27.5% and 27.0% at week 2 post-challenge, respectively, and 27.9% and 25.4% at week 8 post-challenge, respectively; $p = 0.03$ for follicular levels of PGT121$^+$ cells vs. PGT121/DEL$^+$ cells at week 8 post-challenge) (Fig. 4D).

Taken together, these data showed that WT and DEL bNAbs infused early after SHIV$_{AD8-EO}$ infection were detected in LNs as early as 2 weeks after challenge and were maintained throughout week 8. Furthermore, WT and DEL bNAbs had a distinct localization pattern in the LNs, whereby WT bNAbs were evenly distributed between follicular and extrafollicular areas while DEL bNAbs tended to mostly concentrate in the extrafollicular areas.

### DEL bNAb-treated monkeys developed distinct SIV Gag-specific CD8$^+$ T cell responses in peripheral blood

The distinct detection patterns of virus and bNAbs between the monkey groups prompted us to investigate if and how early bNAb infusions after SHIV$_{AD8-EO}$ infection altered the immune cell population dynamics in peripheral blood and LNs. Major cell populations (T cells, B cells, monocytes, and NK cells) and subsets of T cells (CD4$^+$ and CD8$^+$ T cells, LN follicular CD8$^+$ (fCD8$^+$) T cells, and germinal center (GC) T follicular helper (Tfh) cells) were assessed by flow cytometry

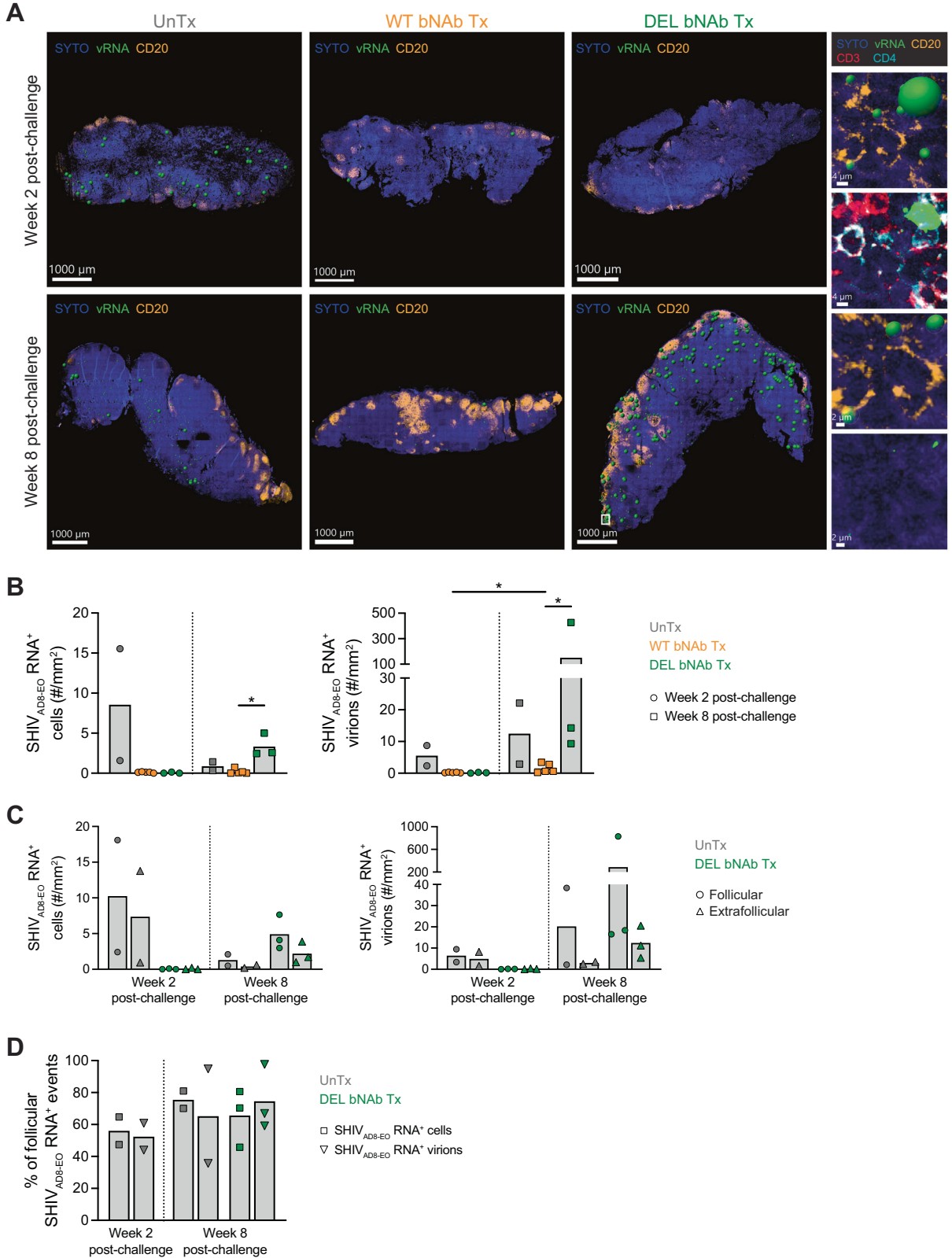

(Supplementary Fig. 5A). Both in peripheral blood and LNs, the levels of all major cell populations remained relatively stable over time, with no apparent differences across monkey groups (Supplementary Fig. 5B). However, as expected, CD4+ T cells decayed (and CD8+ T cells reciprocally increased) later in WT bNAb-treated monkeys than in untreated and DEL bNAb-treated monkeys (Supplementary Fig. 5C), consistent with the longer time it took for plasma viremia to emerge in

the former animals (Fig. 1C). Finally, although levels of fCD8+ T cells and GC Tfh cells appeared to increase in both WT and DEL bNAb-treated monkeys at week 20 post-challenge, they were not significantly different from untreated monkeys (Supplementary Fig. 5D). Taken together, these data indicate that SHIV$_{AD8-EO}$ infection followed by early bNAb treatment affected the dynamics of CD4+ T cells but not of other immune cell subsets in either peripheral blood or LNs.

**Fig. 3 | Detection of SHIV$_{ADS-EO}$ RNA in LN sections. A** Representative RNAscope staining of LN sections from monkeys that were left untreated or were treated with bNAbs early after SHIV$_{ADS-EO}$ challenge. Staining was performed in one LN section per timepoint per animal. Big images (scale bars: 1000 μm) show SHIV$_{ADS-EO}$ RNA (green), CD20 (yellow), and cellular nucleic acids (SYTO, blue) in untreated (left), WT bNAb-treated (middle), and DEL bNAb-treated (right) monkeys at weeks 2 (top) and 8 (bottom) post-challenge. Small images show a LN SHIV$_{ADS-EO}$ RNA$^+$ cell and virions in the DEL bNAb-treated monkey at week 8 post-challenge. Top image (scale bar: 4 μm) shows green spheres denoting the SHIV$_{ADS-EO}$ RNA$^+$ cell and virions; top middle image (scale bar: 4 μm) shows the real signal for the SHIV$_{ADS-EO}$ RNA$^+$ cell; and the middle bottom and bottom images (scale bars: 2 μm) show the spheres and real signal for virions, respectively. Viral events are shown in association with cellular nucleic acids (blue) alone or in combination with either CD20 (yellow) or CD3 (red) and CD4 (turquoise). **B, C** RNAscope quantification of SHIV$_{ADS-EO}$ RNA$^+$ cells (left) and virions (right) in whole LN sections and in follicular (circles) and extra-follicular (triangles) areas from untreated ($n = 2$), WT bNAb-treated ($n = 5$), and DEL bNAb-treated ($n = 3$) monkeys. $P = 0.0357$ for WT vs. DEL group at week 8 (B, left and right) and $p = 0.0238$ for week 2 vs. 8 in WT group (B, right), two-sided p-values. **D** Percentage of follicular SHIV$_{ADS-EO}$ RNA$^+$ cells (squares) and virions (triangles) in untreated ($n = 2$) and DEL bNAb-treated ($n = 3$) monkeys. Bar graphs show the mean and individual datapoints (**B–D**). The Mann-Whitney test was used to detect significant differences in viral parameters between WT and DEL bNAb-treated monkeys at each timepoint (**B**), between timepoints for each bNAb-treated group in whole LN sections (**B**) and in follicular or extrafollicular areas (**C**), and between follicular and extrafollicular areas at each timepoint in DEL bNAb-treated monkeys (**C**). $N$ indicates the number of biological replicates (monkeys). Source data are provided in the Source Data file. Tx treatment, UnTx untreated, vRNA viral RNA.

Next, we evaluated the effect of bNAb therapy initiated during acute SHIV$_{ADS-EO}$ infection on virus-specific CD8$^+$ T cell responses. These responses were evaluated by flow cytometry after 6-hour in vitro stimulation of PBMCs and LN cells with SIV Gag peptide pool and defined as CD8$^+$ T cells expressing CD107a as a marker of degranulation or co-expressing the early activation marker CD69 with IFNγ, MIP-1β, or TNF (Supplementary Fig. 6A). In peripheral blood, SIV Gag-specific CD8$^+$ T cell responses were initially detected in most of the untreated monkeys at week 4 post-challenge, followed by DEL bNAb-treated monkeys at week 6, and finally WT bNAb-treated monkeys at week 16 (Fig. 5A left). Thus, during the first 8 weeks post-challenge, CD8$^+$ T cell responses in WT bNAb-treated monkeys were significantly lower than in untreated monkeys ($p = 0.05$, $p = 0.05$, and $p = 0.005$ at weeks 4, 6, and 8 post-challenge, respectively) and DEL bNAb-treated monkeys ($p = 0.05$ and $p = 0.04$ at weeks 6 and 8 post-challenge, respectively) (Fig. 5A left). A similar pattern was observed in LNs, where most of the SIV Gag-specific CD8$^+$ T cell responses at week 8 post-challenge came from untreated and DEL bNAb-treated monkeys, followed by WT bNAb-treated monkeys at week 20 post-challenge (Fig. 5B right). This detection pattern of virus-specific CD8$^+$ T cell responses in peripheral blood and LNs was consistent with the detection of virus in the plasma and LNs of these animals (Figs. 1C and 3B), showing that virus-specific CD8$^+$ T cell responses tracked with viral antigen levels. In LNs, SIV Gag-specific fCD8$^+$ T cell responses were still very low in DEL bNAb-treated monkeys at week 8 post-challenge ($p = 0.04$ for untreated vs. DEL bNAb-treated monkeys) and were mostly detected in both DEL and WT bNAb-treated monkeys at week 20 post-challenge (Supplementary Fig. 6B). Within each monkey group, the magnitude of SIV Gag-specific CD8$^+$ T cell responses was comparable between peripheral blood and LNs, except for WT bNAb-treated monkeys, whose responses in LNs were significantly higher than in peripheral blood at week 20 post-challenge ($p = 0.003$) (Supplementary Fig. 6C). In addition, the magnitude of the responses did not vary significantly among monkey groups at week 20 post-challenge when responses were measurable in all three monkey groups (Fig. 5A).

To investigate whether the quality of the SIV Gag-specific CD8$^+$ T cell responses at week 20 post-challenge was affected by early bNAb treatment, we conduced polyfunctionality analysis using the Simple Presentation of Incredibly Complex Evaluations (SPICE) software package. In peripheral blood, the polyfunctional profile of those responses significantly differed between WT and DEL bNAb-treated monkeys ($p = 0.04$ for pie chart comparison) (Fig. 5B). DEL bNAb-treated monkeys had significantly higher levels of CD8$^+$ CD69$^+$ T cells producing IFNγ alone than WT bNAb-treated and untreated monkeys ($p = 0.02$ for both WT vs. DEL bNAb treatment and untreated vs. DEL bNAb treatment) (Fig. 5C top). Although DEL bNAb-treated monkeys also had higher levels of IFNγ/MIP-1β double-producing CD8$^+$ CD69$^+$ T cells ($p = 0.001$ for WT vs. DEL bNAb treatment and $p = 0.01$ for untreated vs. DEL bNAb treatment) (Fig. 5C top), the levels were overall too low and likely negligible. In contrast, during and shortly after the DEL bNAb infusions, plasma levels of MIP-1β as measured by Luminex were noteworthy and higher when compared to the other monkey groups ($p = 0.03$ and $p = 0.05$ at day 17 post-challenge and $p = 0.007$ and $p = 0.04$ at week 3 post-challenge for untreated vs. DEL bNAb treatment and WT vs. DEL bNAb treatment, respectively) (Supplementary Fig. 6D, E). The polyfunctional profile of the LN responses was comparable among the monkey groups (Fig. 5B). While a very small population, WT bNAb-treated monkeys had significantly higher levels of MIP-1β/TNF/CD107a triple-expressing LN CD8$^+$ CD69$^+$ T cells when compared to untreated monkeys ($p = 0.03$) (Fig. 5C bottom). Within each monkey group, SIV Gag-specific CD8$^+$ T cell responses significantly differed between peripheral blood and LNs, in that degranulation alone accounted for the vast majority of the CD8$^+$ T cell responses in LNs ($p < 0.001$, $p = 0.002$, and $p < 0.001$, for pie chart comparisons of PBMCs vs. LN cells in untreated, WT bNAb-treated, and DEL bNAb-treated groups, respectively) (Fig. 5B).

In conclusion, these data indicate that although early bNAb administration delayed the development of virus-specific cellular immunity, likely by delaying exposure to viral antigens, circulating virus-specific CD8$^+$ T cell responses developing upon DEL bNAb treatment had a distinct profile when compared to those developing upon WT bNAb treatment or in the absence of treatment.

## LN immune cells from DEL bNAb-treated monkeys had distinct transcriptomic profiles at week 8 post-challenge

Given the presence of infused bNAbs in LNs at weeks 2 and 8 post-challenge (Fig. 4), we next investigated the immunological effects of bNAbs binding to LN cells through their Fc domain. Rhesus macaque monocytes and NK cells express FcγRs[48], so we hypothesized that the bNAbs used in this study could bind to these cell types and alter their transcriptomic profiles. To test this hypothesis, we incubated LN cells from naïve NHPs with each bNAb and assessed binding to monocytes, NK cells, and T cells by flow cytometry. Like in the confocal microscopy experiments (Fig. 4), we used an anti-human immunoglobulin (Ig) kappa light chain antibody to recognize VRC07-523-LS and VRC07-523-LS/DEL and an anti-human Ig lambda light chain antibody to recognize PGT121 and PGT121/DEL. All tested bNAbs bound to LN monocytes and NK cells, and only minimally to T cells (Supplementary Fig. 7). In addition, more monocytes were bound to the DEL bNAbs than to their WT counterparts (Supplementary Fig. 7), consistent with the higher affinity of DEL bNAbs to rhesus FcγRII and FcγRIII[40].

We then FACS-sorted LN monocytes and NK cells (Supplementary Fig. 8A) as well as different subsets of CD4$^+$ and CD8$^+$ T cells (Supplementary Fig. 8B) from untreated and bNAb-treated monkeys (both infected and uninfected) before challenge and 2 and 8 weeks after challenge and conducted transcriptomic analysis using bulk RNA sequencing. T cell sorts were performed on samples from the second batch of monkeys (4 monkeys per group) due to scarcity of LN samples

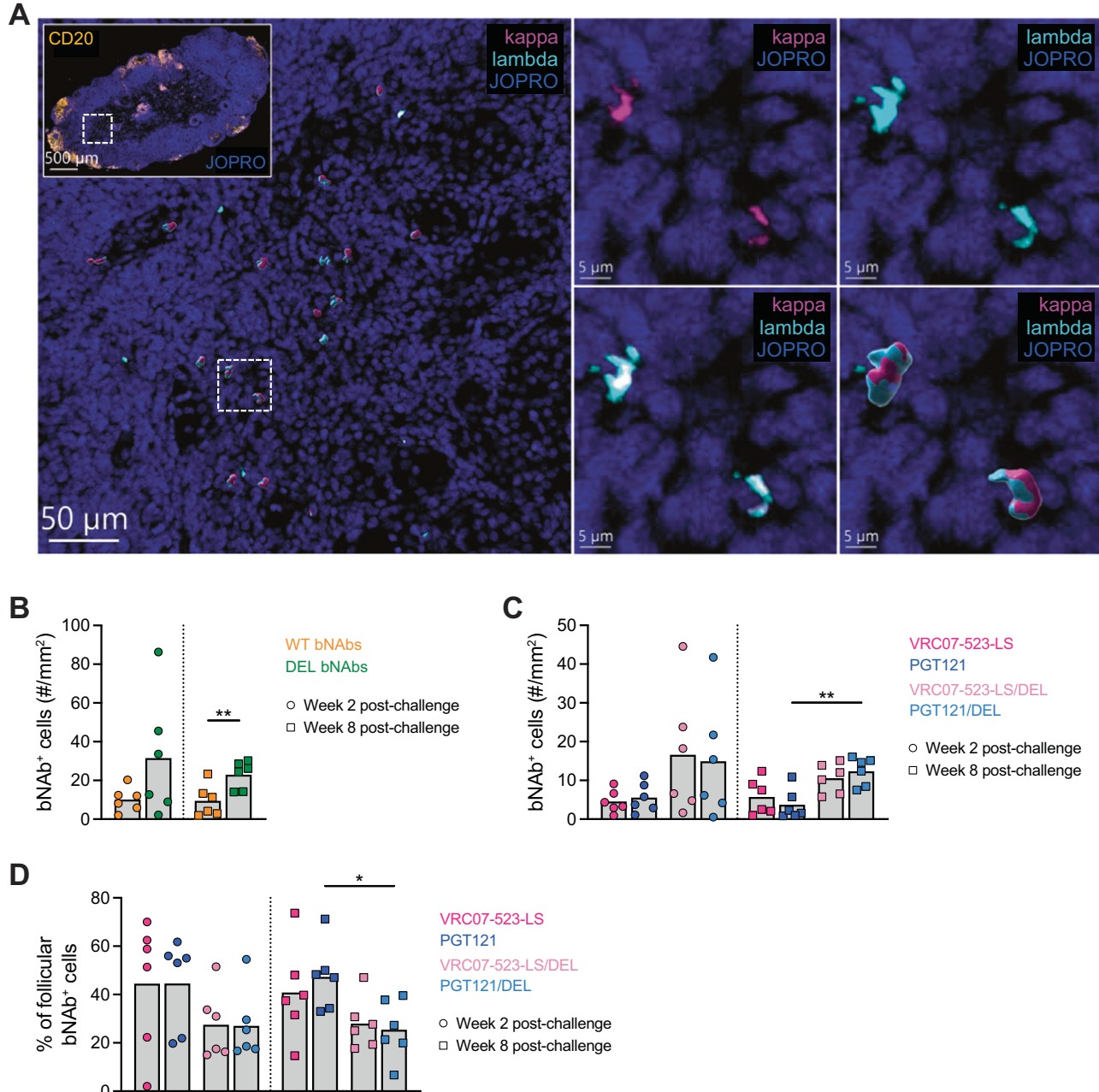

**Fig. 4 | Detection of infused bNAbs in LN sections. A** Representative confocal microscopy staining to detect infused bNAbs in LN sections from monkeys that were treated with bNAbs early after SHIV$_{AD8-EO}$ challenge. bNAb staining was performed in one LN section per timepoint per animal. Inset (scale bar: 500 μm) shows CD20 (yellow) and cellular nucleic acids (JOPRO, blue) in the whole LN section; big image (scale bar: 50 μm) shows human Ig kappa light chain$^+$ cells (pink), human Ig lambda light chain$^+$ cells (turquoise), and cellular nucleic acids (blue); and small images (scale bars: 5 μm) show cellular nucleic acids (blue) with only human Ig kappa light chain$^+$ cells (pink) (top left), only human Ig lambda light chain$^+$ cells (turquoise) (top right), or an overlay of both (bottom). The big image and small image on the bottom right show pink and turquoise Surface objects assigned by Imaris for easier visualization of human Ig kappa light chain$^+$ and lambda light chain$^+$ cells. All other images show real staining signals. Quantification of total WT bNAb$^+$ and total DEL bNAb$^+$ cells (**B**) or VRC07-523-LS$^+$, PGT121$^+$, VRC07-523-LS/

DEL$^+$, and PGT121/DEL$^+$ cells (**C**) in whole LN sections at weeks 2 (circles) and 8 (squares) post-challenge. $N = 6$ per group per timepoint. Each $p = 0.0087$, two-sided $p$-values. **D** Percentage of follicular VRC07-523-LS$^+$, PGT121$^+$, VRC07-523-LS/DEL$^+$, and PGT121/DEL$^+$ cells in LN sections at weeks 2 (circles) and 8 (squares) post-challenge. $N = 6$ per group per timepoint. $P = 0.0260$, two-sided $p$-value. Bar graphs show the mean and individual datapoints (**B**–**D**). The Mann-Whitney test was used to detect significant differences in bNAb$^+$ cell levels at each timepoint or between timepoints for each type of bNAb (WT or DEL) (**B**). The Mann–Whitney test was also used to detect significant differences between the levels of VRC07-523-LS$^+$ and PGT121$^+$ cells, VRC07-523-LS/DEL$^+$ and PGT121/DEL$^+$ cells, VRC07-523-LS$^+$ and VRC07-523-LS/DEL$^+$ cells, or PGT121$^+$ and PGT121/DEL$^+$ cells at each timepoint, or between timepoints for cells bound to each bNAb (**C**, **D**). $N$ indicates the number of biological replicates (monkeys). Source data are provided in the Source Data file. kappa, human Ig kappa light chain; lambda, human Ig lambda light chain.

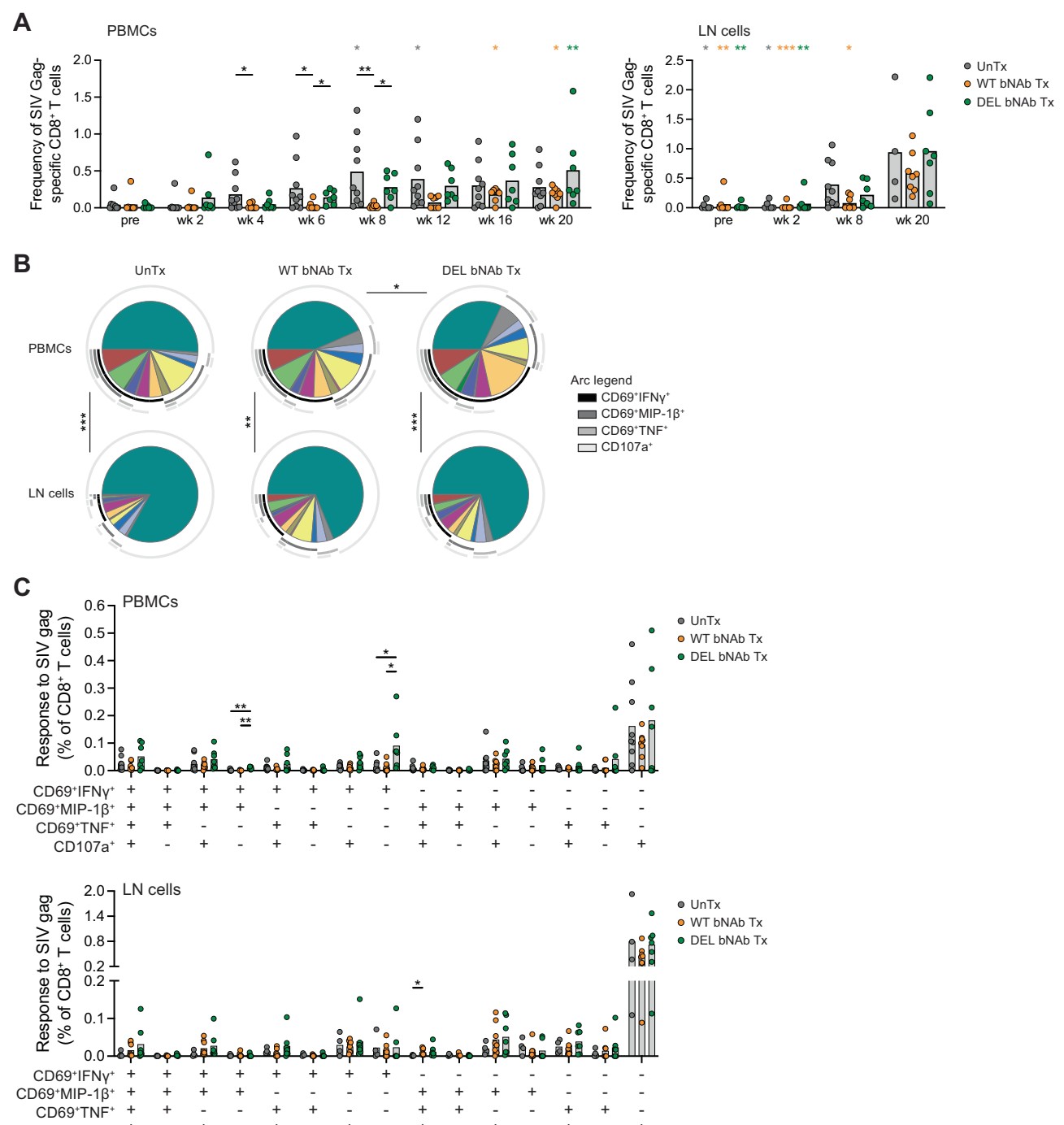

**Fig. 5 | SIV Gag-specific CD8⁺ T cell responses. A** Frequency of SIV Gag-specific CD8⁺ T cells in PBMCs (left) and LN cells (right) from monkeys that were left untreated or were treated with bNAbs early after SHIV_ADS8-EO challenge. $N = 9$ untreated monkeys except for LN cells pre: $n = 7$, week 2: $n = 6$, and week 20: $n = 4$. $N = 8$ and 7 WT and DEL bNAb-treated monkeys, respectively. $P = 0.0474, 0.0480$, and 0.0053 for untreated vs. WT group at week 4, 6, and 8, respectively, and $p = 0.0485$ and 0.0408 for WT vs. DEL group at weeks 6 and 8, respectively (left) (**B**, **C**) Polyfunctional profile of SIV Gag-specific CD8⁺ T cell responses in PBMCs (top) and LN cells (bottom) at week 20 post-challenge. $N = 9$ and 4 untreated monkeys for PBMCs and LN cells, respectively; $n = 8$ and 7 WT and DEL bNAb-treated monkeys, respectively. **B** $P = 0.04$ for WT vs. DEL group in PBMCs; $p < 0.001$ for PBMCs vs. LN cells in untreated and DEL groups, and $p = 0.002$ for PBMCs vs. LN cells in WT group; two-sided $p$-values. **C** PBMCs, IFNγ⁺MIP-1β⁺TNF⁻CD107a⁻ CD8⁺ T cells: $p = 0.0099$ and 0.0014 for untreated vs. DEL and WT vs. DEL group, respectively. PBMCs, IFNγ⁺MIP-1β⁻TNF⁻CD107a⁻ CD8⁺ T cells: $p = 0.0209$ and 0.0240

for untreated vs. DEL and WT vs. DEL group, respectively. LN cells, IFNγ⁻MIP-1β⁺TNF⁺CD107a⁺ CD8⁺ T cells: $p = 0.0333$ for untreated vs. WT group. Graphs show the mean (**A**–**C**) and individual datapoints. The Kruskal–Wallis test followed by Dunn's multiple comparison test was used to detect significant differences between monkey groups at each timepoint (**A**) and for each functional readout combination (**C**), and between timepoints for each monkey group (A, colored statistics). Statistics in gray, orange, and green, denote, respectively, differences from week 2 for untreated monkeys, week 4 for WT bNAb-treated monkeys, and pre-challenge for DEL bNAb-treated monkeys in PBMCs (A, left), and from week 20 in LN cells (A, right). The permutation test was used to compare pie charts (**B**). $N$ indicates the number of biological replicates (monkeys). Source data are provided in the Source Data file. LN lymph node, PBMCs peripheral blood mononuclear cells, pre pre-challenge, Tx treatment, UnTx untreated, wk week.

from the first batch of monkeys (6 monkeys per group) (Fig. 1). Gene set enrichment analysis (GSEA) revealed that LN monocytes and NK cells from treated monkeys significantly upregulated antiviral IFN-α signaling at week 8 post-challenge, but not earlier in infection, when compared to the pre-challenge timepoint (Fig. 6, A and B). At week 2 post-challenge, monocytes from DEL bNAb-treated monkeys and NK cells from both WT and DEL bNAb-treated monkeys significantly upregulated TNF signaling via NF-κB (Fig. 6A, B and Supplementary Fig. 9A, B). However, such upregulation only persisted at week 8 in DEL bNAb-treated monkeys, whereas it was absent from WT bNAb-treated and untreated monkeys at the same timepoint (Fig. 6A, B and Supplementary Fig. 9A, B). Also at week 8 post-challenge, IL-6/JAK/ STAT3 signaling was upregulated exclusively in monocytes from DEL bNAb-treated monkeys (Fig. 6A). Taken together, these data indicate that at week 8 post-challenge, only DEL bNAb treatment induced proinflammatory signatures in LN monocytes and NK cells.

Although bNAbs only minimally bound to T cells (Supplementary Fig. 7), we hypothesized that, at week 8 post-challenge, T cells could be indirectly affected by DEL bNAbs downstream of monocyte engagement. To assess the effect of DEL bNAbs on LN T cells at week 8 post-challenge without having viremia as a confounding factor, we compared the transcriptomic programs of T cells from DEL bNAb-treated monkeys at week 8 post-challenge to those from untreated monkeys at week 2 post-challenge since their viral loads were comparable. Samples from DEL bNAb-treated monkeys at week 2 post-challenge were used to assess the effect of DEL bNAbs longitudinally; however, monkeys were still aviremic at that timepoint. GSEA on the T cell transcriptomic dataset of untreated and DEL bNAb-treated monkeys revealed that antiviral IFN-α signatures were upregulated in all studied T cell types only when the virus was already detectable in plasma and LNs (i.e., at week 2 post-challenge in untreated monkeys and at week 8, but not week 2, post-challenge in DEL bNAb-treated monkeys) (Fig. 6C). Interestingly, exclusively in DEL bNAb-treated monkeys at week 8 post-challenge, cell cycle-related genes were significantly upregulated in most non-naïve CD4[+] and CD8[+] T cell subsets (Fig. 6C and Supplementary Fig. 10A), and IL-2/STAT5 signaling and TNF signaling via NF-κB were significantly downregulated in virtually all T cell subsets (Fig. 6C and Supplementary Fig. 10B). These expression patterns were not only observed when comparing DEL bNAb-treated monkeys between the timepoint of detectable virus (week 8 post-challenge) and no detectable virus (week 2 post-challenge), but also when comparing timepoints of detectable virus between DEL bNAb-treated and untreated monkeys (weeks 8 and 2 post-challenge, respectively) (Fig. 6C). This implicates DEL bNAb treatment, and not viremia, as the main driver of these expression patterns. In conclusion, LN immune cells of DEL bNAb-treated monkeys showed distinct transcriptomic profiles at week 8 post-challenge.

## Discussion

In recent years, potent anti-HIV-1 bNAbs have been discovered and extensively characterized. Not only can bNAbs neutralize the virus through their Fab domain, but they also have the potential to trigger various host antiviral effector functions through their Fc domain, which makes them promising candidates for HIV-1 prevention and treatment strategies. Although anti-HIV-1 bNAbs with enhanced or ablated Fc-mediated effector functions have been infused in NHPs during chronic SHIV infection[39–41], administration of such bNAbs in the acute phase of SHIV or HIV infection has not been explored. In this study, we investigated the virological and immunological effects of therapeutically administering to NHPs the anti-HIV-1 bNAbs VRC07-523-LS and PGT121 either in their WT conformation or with the DEL mutation, which increases binding affinity to rhesus FcγRIII and FcγRII and Fc-mediated effector functions in vitro[40,42]. bNAbs were administered to NHPs very early after SHIV_{AD8-EO} infection, specifically at days 3, 10, and 17 post-challenge. Early triple administration of WT or DEL

bNAbs delayed the emergence of virus in both plasma and LNs as opposed to untreated monkeys. This delay was shorter when DEL bNAbs were administered when compared to WT bNAbs, consistent with the shorter duration of DEL bNAbs in circulation, and viremia was eventually apparent in all monkey groups. Circulating virus-specific CD8[+] T cell responses developing thereafter had a distinct polyfunctional profile in DEL bNAb-treated monkeys, characterized by higher frequencies of CD8[+] CD69[+] T cells producing IFNγ alone when compared to WT bNAb-treated and untreated monkeys. In LNs, the presence of both WT and DEL bNAbs was demonstrable as early as 2 weeks after challenge and was sustained up to week 8, but while WT bNAbs were similarly distributed between follicular and extrafollicular areas, DEL bNAbs mostly bound to cells in the extrafollicular areas. In DEL bNAb-treated monkeys at week 8 post-challenge, LN monocytes and NK cells upregulated proinflammatory signaling pathways and LN T cells downregulated NF-κB signaling in response to TNF. Overall, early administration of bNAbs with increased affinity to FcγRs affected both innate and adaptive cellular immunity. The use of such bNAbs should be considered when designing future passive bNAb therapies aimed at harnessing different components of the immune system and modulating host immune responses.

To our knowledge, this is the first study where anti-HIV-1 bNAbs with increased affinity to FcγRs were administered in the acute phase of SHIV infection and where the presence, localization, and cellular effects of infused bNAbs were investigated in detail in the LNs of treated monkeys. The early bNAb therapy strategy followed in this study consisted of triple administration of a combination of two anti-HIV-1 bNAbs starting at day 3 post-intrarectal SHIV_{AD8-EO} challenge and resulted in delayed emergence of measurable virus in both plasma and LNs. According to our previous study where we identified the Fiebig-equivalent stages of SHIV_{AD8-EO} infection[49], these findings mean that plasma and LN viremia can be successfully delayed if administration of bNAbs starts in the eclipse phase of infection. Our early bNAb strategy differs from that used by Bolton et al.[28], where monkeys received either WT VRC07-523 and PGT121 or WT VRC01 on day 10 after SHIV_{SF162P3} challenge, followed by daily ART initiated 11 days after bNAb infusion. In that study, bNAbs were administered when viremia was already detectable in plasma, and although they lowered peak viremia, they did so to a similar extent than daily ART initiated on day 10 post-SHIV_{SF162P3} challenge in another group of monkeys[28]. Moreover, bNAb administration did not prevent viral rebound post-ART interruption[28]. In our study, plasma virus eventually emerged in treated monkeys as strong ADA responses against the infused bNAbs developed and plasma bNAb levels declined. Of note, ADA responses against the DEL bNAbs were stronger than those against the WT bNAbs. ADA responses occurred in NHPs since bNAbs from a different species (human) were used and were also reported following infusion of PGT121 and N6-LS in NHPs[50]; however, they have seldom been detected in human clinical trials that used human bNAbs[23,51–53]. At the time of peak viremia, we found no evidence of viral escape mutations to the infused bNAbs. This could be due to the use of two bNAbs targeting distinct sites of HIV-1 Env, which decreased selective pressure on the viral population, and/or the limited time that bNAbs were maintained in circulation (5–12 weeks post-challenge for WT bNAbs and 2–9 weeks post-challenge for DEL bNAbs).

Detailed investigation of virus-specific CD8[+] T cell responses at week 20 post-challenge, when plasma virus was detectable in all animal groups and responses were similar in magnitude across groups, revealed a distinct response profile in peripheral blood of DEL bNAb-treated monkeys when compared to WT bNAb-treated and untreated monkeys. This profile was set apart for its significantly higher levels of SIV Gag-specific CD8[+] CD69[+] T cells producing IFNγ alone. Similar viral burdens among monkey groups rules out an effect of viremia alone in the different response patterns detected, and one can hypothesize that DEL bNAbs had a direct effect on FcγR-expressing

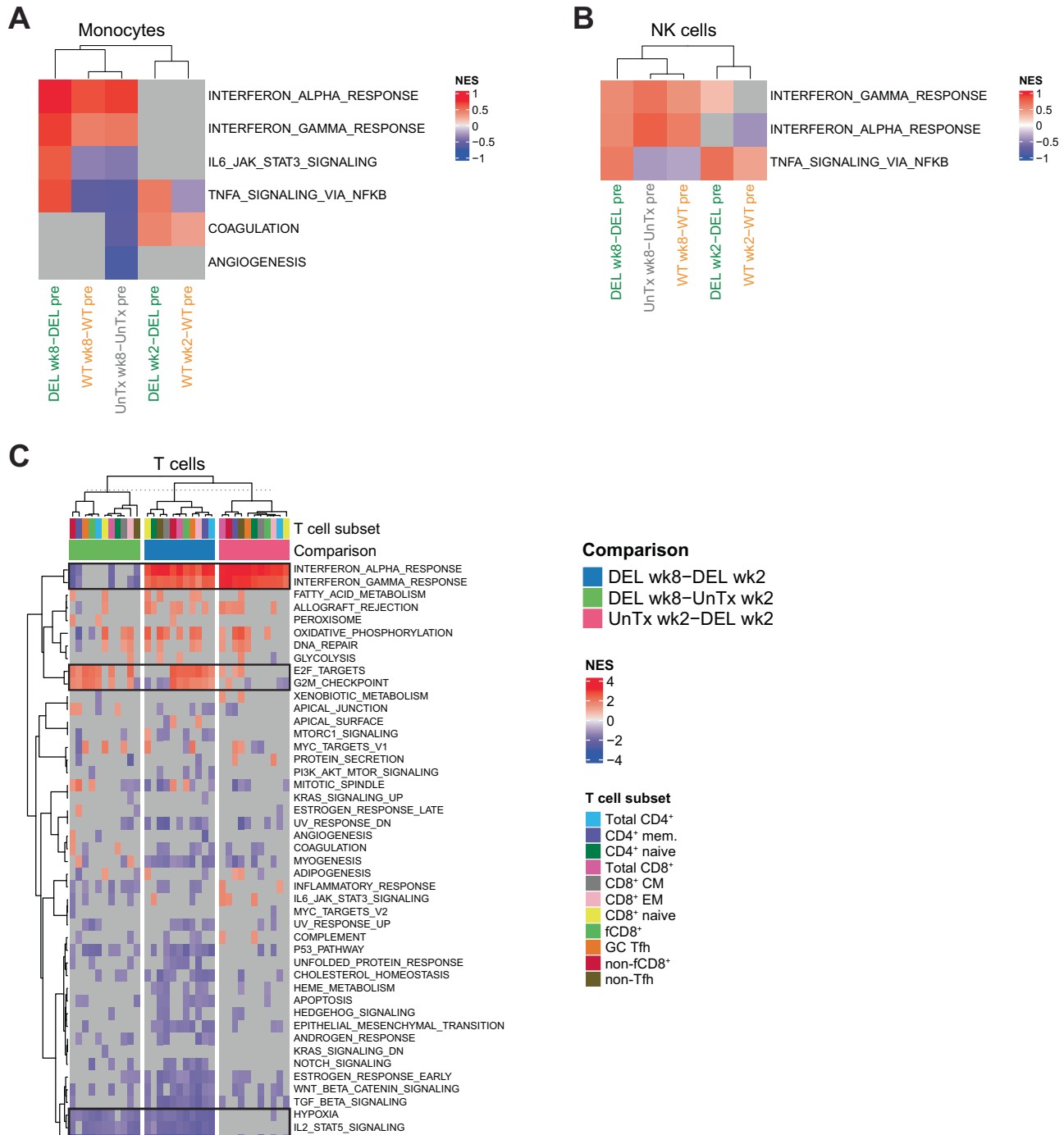

**Fig. 6 | Transcriptomic programs in LN cells from SHIV$_{AD8-EO}$-challenged monkeys on or off early bNAb therapy.** Heatmaps showing the modulation of pathways in sorted LN monocytes (**A**), NK cells (**B**), and T cell subsets (**C**) from untreated and bNAb-treated monkeys at week 2 or 8 post-SHIV$_{AD8-EO}$ challenge when compared to pre-challenge or week 2 post-challenge, as determined by transcriptomic analysis. Monkeys were either left untreated or were treated at days 3, 10, and 17 post-SHIV$_{AD8-EO}$ challenge with either VRC07-523-LS and PGT121 or VRC07-523-LS/DEL and PGT121/DEL. Pathways are indicated in rows (database: MSigDB Hallmark) and the comparisons of interest (monkey group & timepoint 1 − monkey group & timepoint 2) in columns. Red cells indicate significant enrichment of a pathway among genes induced in monkey group & timepoint 1 (GSEA nominal

$p \leq 0.05$ and NES > 0); blue cells indicate significant enrichment of a pathway among genes downregulated in monkey group & timepoint 1 (GSEA nominal $p \leq 0.05$ and NES < 0); and gray cells correspond to non-significant enrichment in monkey group & timepoint 1 (GSEA nominal $p > 0.05$), when compared to monkey group & timepoint 2. A two-sided permutation test was used to calculate the nominal $p$-value and the Benjamini-Hochberg procedure was used to calculate the adjusted $p$-value (**A**–**C**). Source data are provided in the Source Data file. CM central memory, EM effector memory, fCD8$^+$ follicular CD8$^+$, GC germinal center, mem. memory, NES normalized enrichment score, pre pre-challenge, Tfh T follicular helper, UnTx untreated, wk week.

antigen-presenting cells (APCs), which ultimately affected CD8[+] T cell responses. In fact, RNA sequencing analyses revealed that in LN monocytes, only DEL bNAb treatment induced proinflammatory signatures characterized by upregulation of TNF signaling via NF-kB at weeks 2 and 8 post-challenge and IL-6/JAK/STAT3 signaling at week 8 post-challenge. In the LNs, it is possible that such proinflammatory signatures detected in monocytes as well as NK cells boosted ADCP and ADCC activities in vivo, similar to what has been demonstrated in vitro for antibodies carrying the DEL mutation[40,42]. However, those effector functions, even if present, were not sufficient to ultimately induce long-term viral control in DEL bNAb-treated monkeys, as discussed below.

The early bNAb therapy regimen employed in our study was based on an earlier study published by Nishimura et al. [29] where administration of WT 3BNC117 and 10–1074 at days 3, 10, and 17 post-intrarectal or intravenous SHIV$_{AD8-EO}$ challenge led to long-term control of infection in approximately half of the monkeys. Despite many similarities between that study and ours in the challenge virus, challenge route, and dose, timing, and route of bNAb infusions, monkeys that received WT bNAbs in our study did not control virus long-term. It is possible that the different stock of SHIV$_{AD8-EO}$ used for challenge and the different bNAbs infused (even though they targeted similar viral epitopes as the ones used by Nishimura et al. [29]) played a role in the different outcome of these studies.

The long-term viral control achieved by Nishimura et al.[29] was interpreted as being mediated by CD8[+] T cells since depletion of this cell type in controller monkeys led to rapid induction of plasma viremia. Thus, this study supported the hypothesis that bNAbs given early in infection before natural antibodies arise may form immune complexes, which, once cross-presented to CD8[+] T cells, may boost antiviral immune responses[30] and ultimately lead to viral control. With the inclusion in our study of a group of monkeys infused with bNAbs carrying the DEL mutation, we aimed to test if increasing the affinity of bNAbs to FcγRs would potentiate the uptake of immune complexes into APCs and induction of downstream antiviral effects. While we demonstrated an effect on APCs and CD8[+] T cells, these monkeys did not control virus long-term. Virus emerged earlier in DEL bNAb-treated monkeys than in WT bNAb-treated monkeys, consistent with the poorer PK profile of DEL bNAbs. This could be due to either the intrinsic nature of DEL bNAbs or their higher affinity to FcγRs, which precluded them from being detected in the plasma. Peak plasma levels of PGT121/DEL were lower (around 2.5-fold) than those of PGT121, and higher affinity of PGT121/DEL to FcγRs with significantly higher levels of LN cells bound to PGT121/DEL than to PGT121 may have contributed to the relatively lower plasma levels of this DEL bNAb. However, VRC07-523-LS/DEL peak plasma levels were substantially lower (around 7.5-fold) than those of its WT counterpart and unlikely to be only explained by the higher affinity of VRC07-523-LS/DEL to FcγRs. Despite the substantially lower plasma levels of VRC07-523-LS/DEL when compared to its WT counterpart, it is noteworthy that VRC07-523-LS/DEL was present between weeks 1 and 3 post-challenge at plasma levels that were, on average, 6- to 13-fold above the IC$_{80}$ and IC$_{50}$ titers for in vitro neutralization of SHIV$_{AD8-EO}$, respectively. In addition, when VRC07-523-LS/DEL was administered to SHIV$_{SF162P3}$-chronically infected monkeys, it led to a decrease in plasma viral loads, despite also being detected in that study at lower levels than its WT counterpart[40]. In our study, DEL bNAbs did not reduce overall viral burden, but the response profile of SIV Gag-specific CD8[+] T cells and the proinflammatory signatures of LN monocytes and NK cells induced in DEL bNAb-treated monkeys but not in untreated monkeys should motivate further research on the cellular effects and mechanisms of action of Fc-modified bNAbs.

Overall, long-term viral control was not achieved in either WT or DEL bNAb-treated monkeys. Importantly, our data suggest that immune complexes may not have formed to the extent required to achieve protection due to a temporal and spatial dissociation between virus and bNAbs in the LNs. In fact, in both treated groups, bNAbs reached the LNs before the virus. Both WT and DEL bNAbs were present in the LNs at weeks 2 and 8 post-challenge, but virus RNA[+] cells and virions were only detected at week 8 post-challenge and only in the DEL bNAb-treated group, reflecting the pattern of virus emergence in the plasma. In addition, when both bNAbs and virus were present in the LNs of DEL bNAb-treated monkeys, their location differed, in that the virus seemed to predominate in the follicles while DEL bNAbs were mostly detected in the extrafollicular areas. Based on these findings, future studies are required to better understand the traffic and dynamics of infused bNAbs in LNs. Matching the emergence and localization of virus and bNAbs in LNs will be crucial to better assess the role of immune complexes in early bNAb therapy strategies.

It is noteworthy that the DEL mutation ablates binding to the complement proteins C1q and C3c[40]. The fact that no difference in long-term viral control was observed in monkeys that received WT or DEL bNAbs after SHIV$_{AD8-EO}$ infection may suggest that complement does not play an important role in the control of infection. Likewise, complement was found to be redundant in antibody-mediated protection of rhesus macaques against SHIV$_{SF162P3}$[36]. Nonetheless, the role of complement in viral control upon early bNAb therapy warrants further focused investigation through the use of bNAb variants that enhance or abolish binding to complement proteins without affecting binding to FcγRs.

Therapeutic strategies initiated in the acute phase of SHIV or HIV infection offer the opportunity to tackle early viral events and potentially change the course of the disease. In this study, we initiated bNAb interventions in monkeys during the eclipse phase of SHIV$_{AD8-EO}$ infection and used two anti-HIV-1 bNAbs, VRC07-523-LS and PGT121, which have already shown to be safe and well tolerated in clinical trials when administered in their WT conformation[51,54]. Monkeys treated early after infection with WT or DEL bNAbs experienced a delay in viremia that was proportional to the duration of the infused bNAbs in plasma. Treatment with DEL bNAbs exerted distinct effects in innate and adaptive immune cells when compared to WT bNAb-treated or untreated monkeys. Overall, our study encourages the design of bNAb strategies that further explore the effects and mechanisms of action of bNAbs with increased affinity to FcγRs.

## Methods
### Study design
All animal procedures and experiments were conducted according to NIH regulations and standards on the humane care and use of laboratory animals as well as the Animal Care and Use Committees of the NIH Vaccine Research Center (VRC) and Bioqual, Inc. (Rockville, MD). Thirty male and female rhesus macaques (*Macaca mulatta*) of Indian origin and aged 2 to 4 years at the time of challenge were housed at Bioqual, Inc. (Rockville, MD). The animals were inoculated intrarectally with 1000 TCID$_{50}$ of SHIV$_{AD8-EO}$[55], as previously described[56]. At days 3, 10, and 17 post-challenge, 20 monkeys were administered intravenously with a combination of either VRC07-523-LS and PGT121 (*n* = 10) or VRC07-523-LS/DEL and PGT121/DEL (*n* = 10), with each bNAb at 10 mg/kg. Ten other monkeys were left untreated. The monkeys did not express the MHC class I alleles *Mamu-A*01*, *Mamu-B*08*, and *Mamu-B*17*, which are associated with spontaneous viral control, and were assigned to each group based on the genotype of their FcγRII and FcγRIII genes (*FCGR2* and *FCGR3*, respectively) for a balanced *FCGR* genotype representation across groups (Supplementary Table 8). For the CD8[+] T cell depletion experiments, some monkeys were intravenously infused once with 50 mg/kg of the anti-CD8β mAb CD8b255R1 (NIH Nonhuman Primate Reagent Resource). Peripheral blood samples and LN biopsies were collected at multiple timepoints after challenge, and plasma, PBMCs, and LN cells were isolated and stored as previously described in ref. 49. Specifically, the

following LNs were sampled at each timepoint: left inguinal LNs before challenge; right inguinal LNs at week 2 post-challenge; left axillary LNs from WT and DEL bNAb-treated monkeys and left or right axillary LNs from untreated monkeys at week 8 post-challenge, and right axillary LNs at week 20 post-challenge.

### Antibodies for infusions

VRC07-523-LS and PGT121 were generated as previously described[43,51]. The LS[45] and DEL[42] mutations were introduced in the heavy chain plasmids by site-directed mutagenesis (GenScript). Antibodies were purified by protein A-Sepharose affinity chromatography (Cytiva). The anti-CD8β mAb CD8b255R1 is a rhesus recombinant IgG1 mAb that was engineered and produced by the Nonhuman Primate Reagent Resource (NIH Nonhuman Primate Reagent Resource Cat# PR-2557, RRID:AB_2716321). All antibody preparations had low endotoxin levels (< 0.1 endotoxin unit/mg).

### *FCGR* genotyping

Total RNA was isolated from fresh, whole NHP blood using the QIAamp RNA Blood Mini Kit (Qiagen) as per the manufacturer's instructions and then reverse transcribed using oligo(dT)$_{20}$ primers (Invitrogen) and SuperScript™ III Reverse Transcriptase (Invitrogen) under the following conditions: 25 °C for 5 min, 50 °C for 1 h, and 70 °C for 15 min. *FCGR* complementary DNA (cDNA) was amplified by PCR using the Platinum™ *Taq* DNA Polymerase High Fidelity (Invitrogen) and previously described primer pairs that allow amplification of rhesus macaque *FCGR2* cDNA (CD32.1 (fw)/CD32.2 (rv) as well as CD32.3 (fw)/CD32.8 (rv)) and *FCGR3* cDNA (FCG3aF (fw)/FCG3aR (rv)) (Sigma)[57]. PCR conditions were as follows: initial denaturation at 94 °C for 2 min followed by 40 cycles of denaturation at 94 °C for 15 s, annealing at 56 °C for 30 s, and extension at 68 °C for 1 min. Final extension occurred at 72 °C for 10 min. The size of amplified *FCGR* fragments was confirmed by gel electrophoresis using 2% agarose gels (Embi Tec). PCR products were sequenced (ACGT, Inc.) and sequencing analysis was performed using Geneious Prime software v.2020.0.5 (Biomatters Ltd.) based on previously reported macaque *FCGR* sequences[58].

### Viral load measurements

Plasma viremia was measured as previously described[59], using well-established qRT-PCR assays that quantify SIV Gag RNA with a detection limit of 15 (standard assay) or 1 (ultrasensitive assay) copy/mL. Cell-associated SIV Gag RNA and DNA were quantified as previously described[60], using snap-frozen dry PBMC pellets obtained right after isolation of PBMCs from peripheral blood. Data from this assay were normalized to $10^6$ cell equivalents based on concurrent quantification of the CCR5 genomic sequence.

### bNAb PK and ADA analyses

Plasma levels of infused bNAbs were measured by ELISA, as previously described[40]. High-binding, 96-well half-area microplates (Corning) were coated for 1 h at 37 °C with 2 µg/mL in phosphate-buffered saline (PBS) of either resurfaced stabilized core 3 (RSC3) or ST09 (manufactured at the VRC, NIAID, NIH), proteins displaying the CD4-binding site and V3-glycan site[9,61] for capture of VRC07-523-LS (or its DEL mutant) and PGT121 (or its DEL mutant), respectively. Plates were then blocked for 1 h at room temperature (RT) with blocking buffer (Tris buffered saline with 2% bovine serum albumin (BSA), 5% skim milk, and 0.1% Tween 20) and then washed in PBS with 0.05% Tween 20. Serial dilutions of heat-inactivated plasma in blocking buffer were added to the plates and incubated for 1 h at RT. Plates were subsequently washed and incubated for 30 min at RT with a horseradish peroxidase (HRP)-conjugated anti-human IgG secondary antibody cross-adsorbed against rhesus IgG (Southern Biotech) to detect bound bNAbs. After washing, binding was visualized with SureBlue 3,3′,5,5′-tetramethylbenzidine microwell substrate (SeraCare). Color reactions were

stopped with 1 N sulfuric acid (Fisher Chemical) and absorbances were measured at 450 nm in a SpectraMax Plus 384 microplate reader equipped with the SoftMax Pro software v.7.1 (Molecular Devices, LLC). Every plate included control plasma samples from naïve monkeys spiked with known concentrations of purified bNAb and 4-parameter logistic standard curves of purified bNAb from which plasma bNAb concentrations were calculated.

ADA responses were measured in a similar ELISA assay, with the exception that plates were initially coated with 2 µg/mL in PBS of purified bNAb and the secondary antibody was an HRP-conjugated anti-monkey IgG with minimal reactivity to human antibodies (Southern Biotech). Data were reported as endpoint titers interpolated from sigmoidal, 4-parameter logistic sample dilution curves at a pre-determined cut point of five times the average absorbance of the blank wells. Every plate included as positive controls standard curves of 5C9 or 9E9 (manufactured at the VRC, NIAID, NIH), anti-idiotype mAbs that recognize CD4-binding site-specific and V3-glycan-specific bNAbs, respectively.

### Full-length SHIV$_{AD8-EO}$ Env deep sequencing

**Sequencing.** RNA was automatically extracted from virions in plasma using RNAdvance Viral Reagent Kit (Beckman Coulter) and epMotion® 5073t (Eppendorf), and immediately reverse transcribed as previously described[46]. Reverse transcription was conducted using the SuperScript IV Reverse Transcriptase (ThermoFisher Scientific) and a reverse transcription primer (CCCGCGTGGCCTCCTGAATTATNNNNNNNNNTRTAA TAAATCCCTTCCAGTCCCCCC) under the following conditions: 50 °C for 10 min, 85 °C for 10 min, and 4 °C hold. The cDNA was treated with proteinase K (Sigma-Aldrich) at 55 °C for 25 min with shaking at 1000 rpm for residual protein removal. The cDNA was then purified with a 2.2:1 volumetric ratio of RNAClean XP solid phase reverse immobilization beads (Beckman Coulter). Copy numbers of resulting cDNAs were determined by limiting dilution PCR using fluorescence-assisted clonal amplification[62]. Purified cDNA molecules were amplified by PCR using the Advantage 2 PCR kit (Takara Bio), forward primer (GAGCAGAAGA-CAGTGGCAATGA), and reverse primer (CCCGCGTGGCCTCCTGAAT TAT), with each primer at a final concentration of 400 nM. PCR conditions were as follows: initial denaturation at 95 °C for 1 min, 31 cycles of denaturation at 95 °C for 10 s, annealing at 64 °C for 30 s, and extension at 68 °C for 7 min, and final extension at 68 °C for 10 min. Amplified DNA products of 3 kb (2.5 kb Env gene) were incorporated into sequencing libraries using the SMRTbell Express Template Prep Kit 2.0 (Pacific Biosciences) and Barcoded Overhang Adapter kit 8A and 8B (Pacific Biosciences) to sequence multiple samples. The libraries were treated by primer annealing and polymerase binding using the Sequel II Binding Kit 2.0 and Internal Control 1.0 (Pacific Biosciences), and then sequenced by a Sequel II system (Pacific Biosciences) with a 20 h movie time under circular consensus sequencing (CCS) mode.

**Data analysis.** For initial data processing, CCS were generated from SMRT sequencing data with minimum predicted accuracy of 0.99 and minimum number of passes of 3 in Pacific Biosciences SMRT Link (v11.0.0.146107)[63]. CCS reads were then demultiplexed using Pacific Biosciences barcode demultiplexer (lima) to identify barcode sequences. The resulting FASTA files were reoriented into 5′-3′ direction using the *vsearch -orient* command in vsearch (v2.21.1). Cutadapt (v4.1) was used to trim forward and reverse primers. Length filtering was performed to remove reads shorter than 2800 nt or longer than 4000 nt. Appropriately sized reads were then binned by their 8-base UMI sequences. UMI bins with 10 or more CCS reads were kept for preliminary analysis. For each bin, reads were clustered with *vsearch -cluster_fast* based on 99% sequence identity. Only bins that yielded a single, predominant cluster (i.e., where the largest cluster was 1) comprised by at least half of the bin's reads and 2) at least twice as large as the second largest cluster) with at least 10 CCS reads were kept.

The cluster consensus sequence generated by the *vsearch -cluster fast* was then used as a reference to map the cluster's reads with *minimap2* (v2.24). The commands *bcftools mpileup -X pacbio-ccs* and *bcftools consensus* were used to determine the final consensus sequences for each bin. Consensus sequences were filtered by searching the BLAST nt database, and non-SHIV sequences thus identified were discarded.

For final single-genome sequence (SGS) determination, putative fake UMI bins (bins that arise due to PCR and/or sequencing errors) were identified and removed with a network approach as previously described in ref. 46. Given two distinct bins $a$ and $b$ with read counts $n_a$ and $n_b$, respectively (assume $n_a \geq n_b$), $a$ and $b$ are connected by an edge if they have edit distance 1 and satisfy the following count criterion: $n_a \geq 2n_b - 1$. To resolve the network formed above, we applied the *adjacency* method[64], which iteratively consolidates smaller bins into larger bins that meet the above criteria. Despite high CCS read accuracy, some errors may persist, particularly those that may arise during conversion of RNA to cDNA (e.g., reverse transcription errors). As such, variant calling was performed using a model describing technical error rates. Given a reverse transcription error rate $R = 1 \times 10^{-4}$, a target insert length $L$, and a number of recovered SGS sequences $N$, the probability of observing a technical variant with at least $c$ occurrences in the sample can be written as $P(C \geq c) = 1 - \mathrm{Binom}_{CDF}(c|N, R)^L$. To determine a cutoff for variant calling, the smallest value $c_v$ for which $P(C \geq c) < 0.01$ was determined, and the minimum number of occurrences to call a variant was set as $c_v + 1$. Variant calling with this minimum count threshold was performed with a custom Python script included alongside the HT-SGS pipeline. Indels were handled separately; at least three occurrences of the exact same indel were criteria for inclusion as a real variant. Variants not meeting these criteria were reverted to the consensus base. To identify virus haplotypes defined by the data, we took the consensus sequences of all UMI bins and collapsed non-unique sequences. We considered the unique sequences bearing different combination of mutations as individual haplotypes.

## Neutralization assays

The *experimental* neutralizing activity of plasma samples was measured using an established virus neutralization assay that utilizes TZM-bl cells and a luciferase reporter gene, as previously described in ref.[55]. Neutralization curves were fitted to a 5-parameter nonlinear regression analysis, and the neutralizing activity was reported as the reciprocal plasma dilution required to inhibit infection by 50% (50% inhibitory dilution, $ID_{50}$) or 80% (80% inhibitory dilution, $ID_{80}$). The neutralization potency of bNAbs was measured by the same assay and reported as the bNAb concentration required to inhibit infection by 50% (50% inhibitory concentration, $IC_{50}$) or 80% (80% inhibitory concentration, $IC_{80}$). The *predicted* $ID_{50}$ or $ID_{80}$ values of plasma samples were a sum of the *predicted* $ID_{50}$ or $ID_{80}$ values for each of the two bNAbs that the monkeys received. These values were obtained by dividing the plasma bNAb concentration at each timepoint by the $IC_{50}$ or $IC_{80}$ value of the bNAb against each virus.

To summarize the *experimental* neutralizing activity of plasma samples at weeks 64 and 116 post-challenge against all 14 tested viruses, a neutralization score was calculated for each monkey at each timepoint, as previously described in ref.[65]. One point was assigned for *experimental* $ID_{50}$ or $ID_{80}$ values between 40 and 99, two points for values between 100 and 999, and three points for values equal or higher than 1000 for each virus in the multiclade panel. These points were then added together for an overall neutralization score.

## Imaging

For imaging studies, LN biopsies had residual fat removed and were fixed overnight at RT in PBS with 4% paraformaldehyde (PFA). LNs were then embedded in paraffin blocks and 5 μm-thick tissue sections were mounted on Superfrost slides (ThermoFisher Scientific).

To detect $SHIV_{AD8-EO}$ RNA in LN sections, RNAscope™ in situ hybridization (Advanced Cell Diagnostics) was performed as previously described[49]. Following imaging, each $SHIV_{AD8-EO}$ RNA⁺ cell and virion were identified and marked in the LN sections using the Imaris' built-in function Spots. This function first identifies fluorescent signals based on a user-specified diameter and then marks them with a large or small green sphere to facilitate visualization of $SHIV_{AD8-EO}$ RNA⁺ cells and virions, respectively. $SHIV_{AD8-EO}$ RNA⁺ cells were identified based on RNAscope signals measuring 7 μm in diameter. This diameter was based on previous literature[66,67] and the average diameter of $SHIV_{AD8-EO}$ RNA⁺ cells randomly selected from our previous studies. Each LN section was imaged through 5 scanning steps (5 Z-stacks) that allowed us to scan through the tissue section thickness at 5 different focal planes. To accept a 7 μm RNAscope signal as a $SHIV_{AD8-EO}$ RNA⁺ cell, it had to be present on at least 3 focal planes and overlay with nuclear staining. These criteria were manually checked for each individual 7 μm RNA-scope signal. $SHIV_{AD8-EO}$ RNA⁺ virions were identified based on RNA-scope signals measuring ≤ 2 μm in diameter. Given the diameter of HIV viral particles is in the 120 nm range[68], 2 μm was the lowest threshold in the Imaris' Spots function that allowed us to accurately capture all RNAscope signals measuring up to 2 μm. To accept each of these RNAscope signals as a $SHIV_{AD8-EO}$ RNA⁺ virion, it had to be present on at least 3 focal planes. For both large and small RNAscope signals described above, additional background artifacts were manually identified and excluded when the RNAscope signal overlapped with highly saturated nuclear staining fluorescence.

To detect infused bNAbs in LN sections, multiplexed confocal imaging was performed using antibodies that recognize either the kappa or lambda light chain of human antibodies, given that VRC07-523-LS and VRC07-523-LS/DEL express kappa light chains[9,43], and PGT121 and PGT121/DEL express lambda light chains[47]. Slides were baked for 1 h at 60 °C, deparaffinized twice in xylene (2 min each), and then rehydrated in decreasing concentrations of ethanol (100%, 95%, 80%, and 0%; 2 min each). Antigen retrieval was performed in Borg Decloaker, RTU (Biocare Medical) for 15 min at 110 °C using a pressure cooker. Slides were then permeabilized and blocked for 1 h with blocking solution (PBS with 0.3% Triton X-100 and 1% BSA). Primary unconjugated antibody against CD20 (clone L26, eBiosciences) was incubated overnight at 4 °C. After washing steps (PBS 1X three times, 15 min each), slides were incubated with the secondary antibody goat anti-mouse IgG2a (Brilliant Violet 421-conjugated, Biolegend) for 2 h at RT. Following washing steps, slides were blocked with 10% normal mouse/goat serum for 1 h at RT. Directly conjugated antibodies against human Ig lambda light chain (Alexa Fluor 488-conjugated, clone RM127, Novus Biologicals) and human Ig kappa light chain (Alexa Fluor 647-conjugated, clone RM126, Novus Biologicals) were incubated for 2 h at RT followed by additional wash steps. Slides were then counterstained for nuclei with JOPRO (Invitrogen) for 20 min at RT, then mounted with Fluoromount-G (Southern Biotech) and allowed to cure for at least 30 min at 33 °C.

Images were acquired using a Nikon C2 confocal microscope (40X objective, 1.40 NA) and processed with Imaris version 9.9.0 (Bitplane). bNAb⁺ cells were quantified using the Surface Wizard tool within Imaris. Like in the RNAscope analysis described above, 7 μm was assigned as the cell diameter. For each bNAb, the Surface generation process required the following steps: 1) the Surface threshold was adjusted so that while background signal was minimized, the Surface expanded to cover the fluorescent signal; 2) the Surface filter was set so that a number of voxels Img=1 filtered out background Surfaces with a small number of voxels while retaining real Surfaces with a big number of voxels; 3) manual clean-up of artifacts due to (a) tissue autofluorescence; (b) oversaturated signal under multiple optical configurations; (c) signal not associated with nuclear staining; and/or (d) signal not detected in at least 3 confocal optical slices was performed. bNAb⁺ cells were quantified in whole LN sections as well as in

follicular and extrafollicular areas. To quantify events in the follicular areas, defined based on CD20 expression, the *Shortest Distance to Surface* statistics within the Object-Object tool of Imaris was used. The number of events in the extrafollicular areas was obtained by subtracting the number of follicular events from the number of events in the whole LN section. The number of events was normalized to the LN areas (total, follicular, or extrafollicular) to account for differences in tissue size.

## In vitro cell assays

For functional assays aimed at measuring SIV Gag-specific CD8+ T cell responses, PBMCs and LN cells were thawed and rested for 1 h at 37 °C / 5% $CO_2$ in complete medium consisting of RPMI-1640 medium supplemented with 10% fetal bovine serum (FBS), 2 mM L-glutamine, 100 µg/mL streptomycin, and 100 U/mL penicillin (all from Gibco). Following incubation, cells were handled as previously described[49]. Briefly, cells were incubated for 6 h with either 4 µg/mL SIV Gag peptide pool (NIH AIDS Reagent Program) or DMSO (unstimulated negative control), along with brefeldin A (Sigma), monensin (GolgiStop, BD Biosciences), and anti-CD107a antibody (clone H4A3, Biolegend) to assess degranulation. Cells were then stained with flow cytometry antibodies to measure their responses. Response levels of SIV Gag-stimulated cells were background subtracted using the residual response levels of unstimulated cells.

To assess binding of bNAbs to LN cells in vitro, cells were thawed in complete medium and incubated for 15 min at RT with 50 µg/mL VRC07-523-LS, VRC07-523-LS/DEL, PGT121, or PGT121/DEL (in vitro bNAb coating experiment). Cells incubated with PBS were used as negative control. Cells were then washed in FACS buffer (PBS with 2% FBS) and surface stained with flow cytometry antibodies.

## Flow cytometry

For immunophenotypic analyses and following the in vitro bNAb coating experiments, cells were stained as previously described in ref.[49] with Zombie UV amine-reactive viability dye (Biolegend), followed by primary conjugated antibodies to surface markers for 15 min at RT. Cells were then washed in FACS buffer and fixed in 1% PFA prior to data acquisition. For functional analyses, cells were stained with Aqua amine-reactive viability dye (Invitrogen) for 10 min at RT, washed in FACS buffer, surface stained with primary conjugated and biotinylated antibodies for 15 min at RT, washed, incubated with streptavidin (Brilliant Violet 421-conjugated, Biolegend) for 15 min at RT, and washed again. Cells were then fixed and permeabilized in Cytofix/Cytoperm (BD Biosciences) for 30 min at 4 °C, washed in Perm/Wash buffer (BD Biosciences), and intracellularly stained with primary conjugated antibodies for 30 min at 4 °C. After washing in Perm/Wash buffer, cells were resuspended in FACS buffer for data acquisition. Absolute cell counts using fresh, whole NHP blood were determined using Trucount tubes (BD Biosciences) according to the manufacturer's instructions.

All antibodies are listed in Supplementary Table 9 and were used at predetermined optimal titers. Data were acquired on a modified BD FACSymphony equipped with 355-, 405-, 488-, 532-, and 628-nm lasers (BD Biosciences) and analyzed using FlowJo software v.9.9.6 and v.10.8.1 (BD Biosciences). The SPICE software v.6.0 (Mario Roederer, VRC, NIAID, NIH) was used for polyfunctionality analyses.

## Transcriptomics of LN cells

**Sorting & sequencing.** LN cells were thawed in complete medium and stained for cell sorting. Briefly, cells were stained with Aqua amine-reactive viability dye for 10 min at RT, washed in FACS buffer, and surface stained with primary conjugated antibodies (Supplementary Table 9) for 15 min at RT. After washing, cells were resuspended in complete medium and different cell subsets were sorted into ice-cold FBS using either a modified FACSAria III or a modified FACSymphony

S6 cell sorter equipped with 355-, 405- (FACSAriaIII) or 407- (FACSymphony S6), 488-, 532-, and 628-nm lasers (BD Biosciences). Samples were kept on ice before and right after the sortings. Cells were then pelleted by centrifugation, lysed in RNAzol RT (Molecular Research Center, Inc.), and snap-frozen in dry ice. Cell lysates were stored at −80 °C until further use.

Total RNA was isolated from cell lysates using the RNAzol RT protocol as per the manufacturer's instructions. RNA Illumina-ready libraries were generated using NEBNext Ultra II RNA Prep reagents (New England BioLabs). Using oligo dT beads (Invitrogen), polyadenylated RNA was purified and subsequently fragmented, followed by double-stranded cDNA synthesis, end repair, and adapter ligation. The ligated DNA was then barcoded and amplified by a limited cycle PCR and the libraries were sequenced using a paired-end 150-base protocol on either a HiSeq 4000 or a NovaSeq 6000 (Illumina).

**Data analysis.** Illumina reads were aligned to the *Macaca mulatta* genome (Mmul_10 build 102) using the STAR aligner (version 2.7.10a) and counted using HTSeq (2.0.2). Raw gene counts were then TMM normalized, and edgeR (version 3.38.4) framework was used for differential expression analysis (a generalized linear model was fit to each gene, with bNAb therapy as the independent variable and gene expression as the dependent variable). Genes were ranked based on their signed (1: for genes upregulated and -1: for genes downregulated) $-\log_{10}$ *p*-value of the LR test implemented in edgeR and submitted to pathway enrichment analysis using GSEA (version 4.1.0, pre-ranked mode) and MSigDB (version 7.4). ComplexHeatmap (version 2.12.1) was used for visualization.

## Plasma MIP-1β quantification

Plasma levels of MIP-1β were measured in a bead-based multiplex assay using the Luminex technology (Millipore), according to the manufacturer's instructions.

## Statistical analyses

Statistical analyses of all but two datasets (the polyfunctionality and transcriptomic datasets) were conducted using Prism software v.9 (GraphPad Software, LLC.). In Prism, the Mann–Whitney test was used to compare two groups with unpaired samples; the Kruskal-Wallis test was used to compare more than two groups with unpaired samples and was followed by the Dunn's multiple comparison test when the Kruskal-Wallis *p*-value was significant; and correlations were evaluated using the Spearman's rank correlation test. To compare polyfunctionality data depicted in pie charts, the permutation test was used in SPICE. Two-sided *p*-values lower than 0.05 were considered significant. For gene set enrichment analyses performed on the transcriptomic data, a two-sided permutation test was used followed by a Benjamini-Hochberg correction to adjust for multiple testing and an adjusted *p*-value equal or lower than 0.05 was considered significant.

## Reporting summary

Further information on research design is available in the Nature Portfolio Reporting Summary linked to this article.

# Data availability

The raw SHIV_{AD8-EO} Env PacBio CCS sequencing data generated in this study have been deposited in the NCBI SRA database under the BioProject accession number PRJNA957445. The raw LN RNA sequencing data generated in this study have been deposited in the NCBI GEO database under the accession number GSE254837. Source data are provided with this paper.

# Code availability

The bioinformatics analysis pipeline developed for SHIV_{AD8-EO} Env deep sequencing is available at https://github.com/niaid/UMI-pacbio-

pipeline/releases[69]. The bioinformatics analysis pipelines used for preprocessing and downstream analyses of the LN transcriptomic data are available at https://github.com/sekalylab/mRNASeq[70] and https://github.com/sekalylab/joana[71], respectively.

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

## Acknowledgements

This work was supported by the Bill and Melinda Gates Foundation (Grant OPP1147555 to R.A.K) and by the Intramural Research Program of the VRC, NIAID, NIH, and in part with federal funds from the National Cancer Institute, NIH, under Contract No. 75N91019D00024/HHSN261201500003I (J.D.L). We thank Saran Bao, Jumagul Noor, John Graves, Caitlyn Miller, Alida Taylor, Hana Bao, Elizabeth McCarthy, Diana Scorpio, and Ruth Woodward from the Translational Research Program, VRC, for help with the animal experiments; Amy Noe, Shing-Fen Kao, Nadesh Nji, Dillon Flebbe, Evan Lamb, and Kathryn Foulds from the NHP Immunogenicity Core, VRC, for processing some NHP samples; Jesmine Roberts-Torres from the Human Immunology Section, VRC, for helping process samples for transcriptomic analyses; Richard Nguyen, Esther Thang, Daniela Ischiu Gutierrez, Erica Smit, and Steve Perfetto from the Flow Cytometry Core, VRC, for assistance with the flow cytometers; Wanwisa Promsote from the Immunology Laboratory, VRC, for sharing the flow cytometry panel for whole blood cell count, and Christopher Gonelli, Joseph Casazza, and Rodrigo Matus-Nicodemos from the Immunology Laboratory, VRC, for helpful scientific suggestions and discussions. We also thank Kevin Carlton and the staff of the Vaccine Production Program, VRC, for manufacturing some of the bNAbs used in monkey infusions and ELISAs; Baoshan Zhang from the Structural Biology Section, VRC, for providing the 3BNC117 bNAb for neutralization assays, and Krisha McKee from the Humoral Immunology Section, VRC, for help with neutralization data analysis. We are also thankful to Martha Nason from the Biostatistics Research Branch, NIAID, NIH, for expert statistical advice; Reza Sadjadpour from the Laboratory of Molecular Microbiology, NIAID, NIH, for assistance with SHIV$_{AD8-EO}$ Env sequencing data analysis; the staff of the Quantitative Molecular Diagnostics Core of the AIDS and Cancer Virus Program, Leidos Biomedical Research, Inc., Frederick National Laboratory for Cancer Research, for measurements of plasma viral load and cell-associated viral nucleic acids; and Matthew Gastinger from Imaris, an Oxford Instruments Company, for expert advice in imaging analyses. The anti-CD8β mAb CD8b255R1 used in this study was provided by the NIH Nonhuman Primate Reagent Resource (P40 OD028116).

## Author contributions

J.D., M.A., L.G., J.R.M., A.P., and R.A.K. designed the research; J.D., G.F., D.R.A., A.R., F.L., K.M., A.M., S.D.S., J.S., C.A.T., S.H.K., M.E.L., and M.A. performed the experiments; J.D., G.F., S.F., J.H., K.M., S.D.S., J.S., P.E.R., Y.N., J.D.L., and M.A. analyzed the data; X.C. and C.L. prepared the bNAbs used for monkey infusions; Y.N. and M.A.M. generated and provided the SHIV$_{AD8-EO}$ challenge stock; X.C., C.L., and W.S. produced and provided reagents for neutralization assays and/or ELISAs; J.D., A.M., C.A.T., K.E., and M.A. processed NHP samples; J-P.T. coordinated and supervised the animal experiments; J.D., G.F., K.L.B., C.P., E.A.B., N.AD-R., D.C.D., R-P.S., J.D.L., M.A., L.G., A.P., and R.A.K. supervised experiments and analyses; and J.D. and R.A.K. prepared figures and wrote the manuscript. All authors reviewed the manuscript and approved it for publication.

## Funding

## Competing interests

The authors declare no competing interests.

## Additional information

Joana Dias [1], Giulia Fabozzi[2], Slim Fourati [3], Xuejun Chen [4], Cuiping Liu [4], David R. Ambrozak[1], Amy Ransier[5], Farida Laboune[5], Jianfei Hu[5], Wei Shi[4], Kylie March[2], Anna A. Maximova[4], Stephen D. Schmidt[6], Jakob Samsel[1,7], Chloe A. Talana [4], Keenan Ernste [4], Sung Hee Ko [8], Margaret E. Lucas [8], Pierce E. Radecki [8], Kristin L. Boswell[1], Yoshiaki Nishimura [9], John-Paul Todd[10], Malcolm A. Martin [9], Constantinos Petrovas[2], Eli A. Boritz[8], Nicole A. Doria-Rose [6], Daniel C. Douek [5], Rafick-Pierre Sékaly [3], Jeffrey D. Lifson[11], Mangaiarkarasi Asokan[4], Lucio Gama[1], John R. Mascola[12], Amarendra Pegu[4] & Richard A. Koup[1] ✉

[1]Immunology Laboratory, Vaccine Research Center, National Institute of Allergy and Infectious Diseases, National Institutes of Health, Bethesda, MD, USA. [2]Tissue Analysis Core, Vaccine Research Center, National Institute of Allergy and Infectious Diseases, National Institutes of Health, Bethesda, MD, USA. [3]Pathology Advanced Translational Research Unit, Department of Pathology and Laboratory Medicine, School of Medicine, Emory University, Atlanta, GA, USA. [4]Virology Laboratory, Vaccine Research Center, National Institute of Allergy and Infectious Diseases, National Institutes of Health, Bethesda, MD, USA. [5]Human Immunology Section, Vaccine Research Center, National Institute of Allergy and Infectious Diseases, National Institutes of Health, Bethesda, MD, USA. [6]Humoral Immunology Section, Vaccine Research Center, National Institute of Allergy and Infectious Diseases, National Institutes of Health, Bethesda, MD, USA. [7]Institute for Biomedical Sciences, George Washington University, Washington, D.C., USA. [8]Virus Persistence and Dynamics Section, Vaccine Research Center, National Institute of Allergy and Infectious Diseases, National Institutes of Health, Bethesda, MD, USA. [9]Laboratory of Molecular Microbiology, National Institute of Allergy and Infectious Diseases, National Institutes of Health, Bethesda, MD, USA. [10]Translational Research Program, Vaccine Research Center, National Institute of Allergy and Infectious Diseases, National Institutes of Health, Bethesda, MD, USA. [11]AIDS and Cancer Virus Program, Frederick National Laboratory for Cancer Research, Frederick, MD, USA. [12]ModeX Therapeutics, Weston, MA, USA. ✉e-mail: rkoup@mail.nih.gov

