## [Peer Review File · Nature Communications]

Administration of anti-HIV-1 broadly neutralizing monoclonal antibodies with increased affinity to Fcγ receptors during acute SHIVAD8-EO infectionReviewers' Comments:

Reviewer #1:

Remarks to the Author:

Dias et al. conducted a study in rhesus macaques to investigate the effects of administering wild-type (WT) broadly neutralizing antibodies (bNAbs) or bNAbs with a specific mutation (S239D/I332E/A330L, referred to as DEL) during acute SHIVAD8-EO infection. This DEL mutation enhances binding to Fc gamma receptors (FcγRs). The bNAbs used in the study were VRC07-523-LS and PGT121.

The study design involved the administration of WT or DEL bNAbs at days 3, 10, and 17 post-intrarectal SHIVAD8-EO challenge, which mirrors a protocol previously described by Nishimura et al. (Nature, 2017). The aim was to evaluate whether early administration of bNAbs with increased affinity to FcγRs could enhance antiviral effects.

Treatment with bNAbs resulted in delayed virus emergence in plasma and lymph nodes (LNs). Notably, DEL bNAbs exhibited faster clearance from plasma and earlier virus emergence compared to WT bNAbs (week 6 vs. week 10 post-challenge), attributed to the emergence of ADA responses. Monkeys treated with DEL bNAbs showed higher levels of circulating virus-specific CD8⁺ T cells producing single or double cytokines (IFNγ and/or MIP-1b) compared to other groups. However, caution should be exercised when interpreting these data, as the overall frequency of these cells was very low. In LNs, while WT bNAbs were evenly distributed between follicular and extrafollicular areas, DEL bNAbs predominantly localized in the latter. By week 8 post-challenge, LN monocytes and NK cells from DEL bNAb-treated monkeys exhibited upregulation of proinflammatory signaling pathways, while LN T cells downregulated TNF signaling via NF-κB. The authors concluded that bNAbs with enhanced Fc binding influenced innate and adaptive cellular immunity, thereby suggesting potential implications for future passive bNAb therapy strategies.

Overall, the paper is well-structured and the methodology appears robust, contributing significantly to the existing body of knowledge on bNAbs in the context of HIV and SHIV prophylaxis. The analysis of LN localization of both virus and bNAbs, as well as the RNA sequencing analysis performed on cell subsets, is insightful. However, the authors are encouraged to address the identified caveats and suggestions.

Concerns regarding the SHIV RNA detection analysis in Figure 3 are noteworthy.

The analysis was conducted at week 8 post-challenge, when the blood viral load was still undetectable in the plasma of WT bNAbs treated group and the positive RNA signal in LNs was extremely low. A parallel analysis of LNs at a time point corresponding to detectable virus in the plasma of the WT bNAb-treated group (e.g., weeks 10-12) is essential to definitively conclude on virus localization between the follicular and extrafollicular areas. Is the amount and localization of RNA⁺ cells/virions at week 10-12 of the WT Ab-group comparable to what was observed at week 8?

Additionally, further clarification is needed regarding the distinction between RNA⁺ cells and RNA⁺

virions based on spot size assignment. There is no clear demonstration that a larger spot corresponds to a cell unless the authors provide a picture clearly showing a merge of the RNA+ signal with nucleus staining. I suggest the authors exercise caution and either modify the terminology accordingly or specify that the size of the observed signal is consistent with that of a single cell.

Information regarding the number of sections analyzed for such analysis is missing and should be indicated in the figure legend. It seems that only one section per animal has been analyzed (as inferred from the histogram in Figure 3B), which may not be sufficient to account for statistical variability in virus abundance along the LN depth.

Similarly, the statistical analysis performed with only two animals in the control group warrants explanation and validation (Figure 3B and Supplementary Figure 4A).

The same observation applies to confocal microscopy analysis presented in Figure 4.

Did the authors measure cell-associated viral DNA in LNs using PCR? It would be interesting to determine if there is any correlation with the results obtained from in situ hybridization analysis.

Considering the relevance of the RNA sequencing analysis in supporting the authors' conclusions regarding immune response modulation, it would be beneficial to assess the robustness of the findings by applying a more stringent fold-change cut-off (e.g., 1.5).

The authors concluded that the study highlights the usefulness of including bNAbs with increased affinity to FcγRs to modulate the host immune response in future interventions for the treatment of HIV-1 infection. While I agree with the authors that further exploration of the effects and mechanisms of action of such bNAbs should be encouraged, one might wonder about their added value considering that the overall viral burden was not affected and was even worse than when using WT bNAbs. This aspect deserves further discussion.

Minor enhancements include specifying the type/origin of LNs analyzed and providing details on the number of replicates and experiments performed in Table 1.

Reviewer #2:

Remarks to the Author:

Dias et al. have conducted a study in rhesus macaques examining how modifying broadly neutralizing antibodies (bNAbs) to better bind to Fc-gamma receptors with the S239D/I332E/A330L (DEL) mutation set affects early treatment of SHIV-AD8-EO infection with bNAbs and the endogenous immune response of the animals. This study builds on the scholarship surrounding bNAb treatment for SHIV infection in non-human primates (NHPs), specifically treatment during the early phase of infection as opposed to the chronic phase. Specifically, the main experiment of this manuscript was based upon the Nishimura et al. (Nature, 2017) study wherein NHPs were challenged with the same virus and given a combination of the wild-type bNAbs 3BNC117 and 10-1074 at days 3, 10, and 17 post-challenge during acute infection. In Dias et al., the macaques were

challenged with SHIV-AD8-EO and given the antibodies VRC07-523-LS and PGT121 in either their wild-type state or with the DEL mutations. While many studies have also examined the effects of modifying the ability of bNABs to bind to Fc receptors, this study appears to be the first to examine in vivo both modifications of the bNABs and their effects during acute SHIV infection in NHPs.

Unlike in Nishimura et al. (Nature, 2017) study, where approximately half of the animals were able to immunologically control the virus, however, Dias et al. found that none of the macaques treated with the antibody combination had durable virologic control. Interestingly, the macaques given the bNAb combination with the DEL mutation had a higher proportion of SIV Gag-specific, CD69+ CD8 T cells producing IFN γ or IFN γ and MIP-1b. The macaques treated with the DEL mutation also had upregulated proinflammatory signatures in lymph node monocytes and increased plasma MIP-1b expression compared to the sham and wild-type bNAb-treated group. As such, this manuscript contains noteworthy data that is of significance to the study of bNAb treatment in SHIV (and HIV) infection, and I believe that the manuscript should be accepted with some revisions.

I have one major concern regarding this study that I believe should be addressed. The addition of the DEL mutation to VRC07-523-LS resulted in a rather extreme difference in peak plasma concentration from 113 ug/mL with the wild-type to 15 ug/mL with the DEL mutant. While some of this difference could be explained by more efficient binding of the antibody to Fc receptors in the case of the DEL mutant, the authors also presented data showing that, at weeks 2 and 8 post-challenge, there was no significant difference in the number of cells with bound VRC07-523-LS wild-type or DEL in the lymph nodes. The PGT121 peak plasma concentrations were far more reasonable with a difference of 139 ug/mL versus 56 ug/mL in the wild-type versus DEL mutant, and the significant difference in the number of lymph node cells with bound PGT121 is much more convincing for the argument that the difference in plasma antibody concentration is likely owed to the enhanced Fc receptor binding. The nearly log difference in plasma concentration between the two versions of VRC07-523-LS just needs slightly more explanation. The human clinical trial detailed in Julg et al. (Nat Med, 2022), where viremic persons living with HIV were given triple bNAb therapy of PGT121, PGDM1400, and VRC07-523-LS, showed little to no development of resistance to VRC07-523-LS in the individuals' viral populations despite relatively high plasma VRC07-523-LS concentrations while the viruses at the same time had become completely resistant to at least one of the other two antibodies when PGDM1400 and/or PGT121 plasma concentrations dipped below approximately 10 ug/mL. While I acknowledge that this is somewhat of a minor quibble and doesn't have much of an effect on the main conclusions of the paper, the difference in the plasma concentrations of the VRC07-523-LS variants is simply too large to not discuss it somewhat more.

Minor Comments:

1. In Figure 6, Panel C, the image would be slightly more legible if the key for the comparisons was above the NES key and made larger or if the comparison was labelled on the actual heatmap where the color bands are at the bottom of the heatmap.
2. In the legend of Supplemental Figure 1, maybe indicate that "DN" means double-negative with the rest of the abbreviations.
3. In Supplemental Figure 7, Panel B, moving the antibody labels to below the divisions of the graphs instead of just using different symbols would make them easier to interpret and more

legible.

Sincerely,
Victoria Walker-Sperling

Reviewer #3:

Remarks to the Author:

The main message of the manuscript is the effect of two Fc-mutated bNAbs in the shaping of immune responses in the context of acute SHIV infection. However, the results do not formally support this conclusion.

The study presents a significant amount of data, yet a substantial portion of the six main figures focuses on aspects such as viral load, pharmacokinetics, ADA, virus and antibody localization. Regrettably, these data, while interesting, appear disconnected from the primary message of the paper, which aims to elucidate the modulation of immune responses by Fc-engineered antibodies.

The unexpected findings regarding the pharmacokinetics of antibodies are notable. Specifically, the observation that the LS mutant (VRCO7-LS) does not exhibit prolonged serum concentrations compared to the unmutated Fc antibody (PGT121) warrants further investigation. Additionally, the striking influence of DEL mutation on the appearance of ADA is intriguing, yet the paper lacks an explanation for this phenomenon. Worthy of note, the appearance of ADA upon PGT121 and LS-mutated anti-HIV-1 bNAbs has already been reported (Hsu et al, 2021; PLoS Pathog 17(2): e1009339. <https://doi.org/10.1371/journal.ppat.1009339>).

The data pertaining to the shaping of immune response by Fc-mutated bNAbs are insufficient and fail to provide convincing evidence. For instance, the frequency of MIP-1 β expressing cells, which is lower than 0.1%, is practically negligible. To derive more robust conclusions, appropriate controls, such as PBMC activation by an irrelevant peptide, are imperative. Furthermore, the concentrations of MIP-1 β detected in the plasma of DEL-treated animals do not support an antibody-mediated effect on its production by CD8 cells. Notably, MIP-1 β can be secreted by different immune cells and it is mostly detected at weeks 2-3 post-challenge, while MIP-1 β -secreting CD8 T cells are identified at 20 weeks post-challenge. Moreover, the frequency of IFN- γ expressing cells in DEL-treated individuals appears to be both notably low and quite variable, with only a minority of animals exhibiting frequencies higher than those in other groups. It would have been enlightening to establish a relationship between these "responding" animals and their viral loads, as the frequency of SIV Gag-specific CD8 cells appears to correlate more closely with viral rebound than with the antibody treatment itself.

The evidence supporting the effect of DEL treatment in shaping monocyte and NK cell properties is limited. Validating the upregulated pathways at the protein level, such as IL-6 and TNF production by monocytes, would strengthen the conclusions. In addition, the observed enhanced inflammatory response in DEL-treated animals at week 8 post-challenge may be attributed to the

high viral load specifically observed in this group of animals at this timepoint, rather than the antibody treatment. Notably, monocytes secrete strong quantities of IL-6 and TNF upon stimulation by HIV-1 and SIV. Additionally, the speculation regarding DEL-antibodies' retention in LN due to FcγR-expressing cell binding lacks in vivo validation.

The transcriptomic analysis of T cells lacks clarity. The choice of group comparisons is ambiguous, and the purported downregulation of TNF and its association with reduced viral replication contradicts data on LN viral burden at week 8 post-challenge.

The discussion is based on speculation, proposing hypothetical modulation of immune responses through bNAb treatment in an Fc-dependent manner, without direct experimental support.

In summary, despite the extensive dataset, the results do not sufficiently support the conclusion that Fc-engineered antibodies modulate adaptive and innate immune responses in SHIV-infected animals. Moreover, the lack of efficient viral control by the bNAbs used in this study raises doubts regarding their potential to modulate antiviral host immune responses effectively.

Reviewer #4:

Remarks to the Author:

I co-reviewed this manuscript with one of the reviewers who provided the listed reports. This is part of the Nature Communications initiative to facilitate training in peer review and to provide appropriate recognition for Early Career Researchers who co-review manuscripts

Reviewer #1

General comments:

“Dias et al. conducted a study in rhesus macaques to investigate the effects of administering wild-type (WT) broadly neutralizing antibodies (bNAbs) or bNAbs with a specific mutation (S239D/I332E/A330L, referred to as DEL) during acute SHIVAD8-EO infection. This DEL mutation enhances binding to Fc gamma receptors (FcγRs). The bNAbs used in the study were VRC07-523-LS and PGT121.

The study design involved the administration of WT or DEL bNAbs at days 3, 10, and 17 post-intrarectal SHIVAD8-EO challenge, which mirrors a protocol previously described by Nishimura et al. (Nature, 2017). The aim was to evaluate whether early administration of bNAbs with increased affinity to FcγRs could enhance antiviral effects.

Treatment with bNAbs resulted in delayed virus emergence in plasma and lymph nodes (LNs). Notably, DEL bNAbs exhibited faster clearance from plasma and earlier virus emergence compared to WT bNAbs (week 6 vs. week 10 post-challenge), attributed to the emergence of ADA responses. Monkeys treated with DEL bNAbs showed higher levels of circulating virus-specific CD8+ T cells producing single or double cytokines (IFNγ and/or MIP-1b) compared to other groups. However, caution should be exercised when interpreting these data, as the overall frequency of these cells was very low. In LNs, while WT bNAbs were evenly distributed between follicular and extrafollicular areas, DEL bNAbs predominantly localized in the latter. By week 8 post-challenge, LN monocytes and NK cells from DEL bNAb-treated monkeys exhibited upregulation of proinflammatory signaling pathways, while LN T cells downregulated TNF signaling via NF-κB.

The authors concluded that bNAbs with enhanced Fc binding influenced innate and adaptive cellular immunity, thereby suggesting potential implications for future passive bNAb therapy strategies.

Overall, the paper is well-structured and the methodology appears robust, contributing significantly to the existing body of knowledge on bNAbs in the context of HIV and SHIV prophylaxis. The analysis of LN localization of both virus and bNAbs, as well as the RNA sequencing analysis performed on cell subsets, is insightful. However, the authors are encouraged to address the identified caveats and suggestions.”

Response:

We thank the reviewer for the positive assessment of our manuscript.

Specific comment 1:

“Concerns regarding the SHIV RNA detection analysis in Figure 3 are noteworthy. The analysis was conducted at week 8 post-challenge, when the blood viral load was still undetectable in the plasma of WT bNAbs treated group and the positive RNA signal in LNs was extremely low. A parallel analysis of LNs at a time point corresponding to detectable virus in the plasma of the WT bNAb-treated group (e.g., weeks 10-12) is essential to definitively conclude on virus localization between the follicular and extrafollicular areas. Is the amount and localization of RNA+ cells/virions at week 10-12 of the WT Ab-group comparable to what was observed at week 8?”

Response:

We thank the reviewer for this comment. We agree with the reviewer that viral loads in the WT bNAb-treated monkeys at week 8 post-challenge are too low to perform any LN virus localization analysis, and for that reason, we had only graphed the proportion of follicular SHIV_{AD8-EO} RNA⁺ events in the untreated and DEL bNAb-treated monkeys (Figure 3D). In the revised version of the manuscript, we have also now

excluded the follicular and extrafollicular data from the WT bNAb-treated monkeys in Figure 3C. Both the figure 3 legend (page 54) and text in the Results (page 14, lines 272-273) were updated accordingly. We agree with the reviewer that week 10 to 12 timepoints would have been optimal to assess virus localization in the LNs of WT bNAb-treated monkeys; however, LN samples were not collected at those timepoints, so we have no way to generate the requested data.

Specific comment 2:

“Additionally, further clarification is needed regarding the distinction between RNA+ cells and RNA+ virions based on spot size assignment. There is no clear demonstration that a larger spot corresponds to a cell unless the authors provide a picture clearly showing a merge of the RNA+ signal with nucleus staining. I suggest the authors exercise caution and either modify the terminology accordingly or specify that the size of the observed signal is consistent with that of a single cell.”

Response:

We thank the reviewer for this comment. Each SHIV_{AD8-EO} RNA⁺ cell and virion were identified and marked in the lymph node (LN) sections using the Imaris’ built-in function Spots. This function first identifies fluorescent signals based on a user-specified diameter and then marks them with a large or small green sphere to facilitate visualization of SHIV_{AD8-EO} RNA⁺ cells and virions, respectively, in standard size pictures, such as the ones displayed in Figure 3A.

SHIV_{AD8-EO} RNA⁺ cells were identified based on RNAscope signals measuring 7 μm in diameter. This diameter was based on previous literature (Cowell et al., *Vet Clin North Am Small Anim Pract*, 2003, PMID: 12512376; Tasnim et al., *Front Immunol*, 2018, PMID: 30093900) and the average diameter of SHIV_{AD8-EO} RNA⁺ cells randomly selected from our previous studies. Each LN section was imaged through 5 scanning steps (5 Z-stacks) that allowed us to scan through the tissue section thickness at 5 different focal planes. To accept a 7 μm RNAscope signal as a SHIV_{AD8-EO} RNA⁺ cell, it had to be present on at least 3 focal planes and overlay with nuclear staining, as demonstrated in Figure R1. These criteria were manually checked for each individual 7 μm RNAscope signal.

Figure R1. Representative example of the overlay between nuclear staining and RNAscope signal in a LN section. The left image shows cellular nucleic acids (SYTO, blue), the middle image shows SHIV_{AD8-EO} RNA (green), and the right image shows both cellular nucleic acids (SYTO, blue) and SHIV_{AD8-EO} RNA (green). The big RNAscope signal indicated by a pink arrow on the bottom right overlays with cellular nucleic acids and denotes a SHIV_{AD8-EO} RNA⁺ cell. All images have a scale bar of 3 μm .

SHIV_{AD8-EO} RNA⁺ virions were identified based on RNAscope signals measuring $\leq 2 \mu\text{m}$ in diameter. Given the diameter of HIV viral particles is in the 120 nm range (Kuznetsov et al., J Virol, 2003, PMID: 14581526), $2 \mu\text{m}$ was the lowest threshold in the Imaris' Spots function that allowed us to accurately capture all RNAscope signals measuring up to $2 \mu\text{m}$. To accept each of these RNAscope signals as a SHIV_{AD8-EO} RNA⁺ virion, it had to be present on at least 3 focal planes.

For both large and small RNAscope signals described above, additional background artifacts were manually identified and excluded when the RNAscope signal overlapped with highly saturated nuclear staining fluorescence.

Considering the important comment from the reviewer, we added the detailed information above on how the RNAscope analysis was performed in the revised version of the manuscript (pages 35-36, lines 759-776).

Specific comment 3:

"Information regarding the number of sections analyzed for such analysis is missing and should be indicated in the figure legend. It seems that only one section per animal has been analyzed (as inferred from the histogram in Figure 3B), which may not be sufficient to account for statistical variability in virus abundance along the LN depth.

Similarly, the statistical analysis performed with only two animals in the control group warrants explanation and validation (Figure 3B and Supplementary Figure 4A).

The same observation applies to confocal microscopy analysis presented in Figure 4."

Response:

We confirm that only one section per timepoint per animal was imaged for SHIV_{AD8-EO} RNA (Figure 3) and bNAbs (Figure 4), and we included this information in the legends of Figures 3 and 4. In our imaging experiments, we have established multiple criteria for cutting appropriate, good quality lymph node (LN) sections and for imaging that take into account the LN depth and we believe address the reviewer's concern:

1. For each biopsy, half of the LN was embedded in a 4% paraformaldehyde-fixed, paraffin-embedded block. Embedding half of the LN (rather than the whole LN) allowed us to slice throughout the depth of the tissue more easily while diverting from the LN edge in which the tissue structure is not well defined.
2. To obtain good quality sections of the half-embedded LN, the first 30-35 μm of tissue were discarded until the tissue structure started to appear. Next, tissue sections of 5 μm each were made, for a total of approximately 30 sections. Tissue sections 1 to 14 were usually used for RNAscope and bNAb panel testing, and the remaining cuts were used for the experiments. For smaller tissues, less tissue was initially discarded and/or fewer sections were made to follow the same sectioning rationale that takes into account the tissue structure. When one section was imaged as part of an experiment (eg. section #16), the contiguous sections (eg. sections #15 and 17) were stored for any repeat experiment and/or to answer follow up questions related to the results obtained from the initial section (section #16).
3. Each 5 μm LN section was imaged using 5 scanning steps (5 Z-stacks), which allowed us to scan throughout the section thickness at 5 different focal planes. Among other acceptance criteria

detailed in the Methods section (pages 35-37), fluorescent signals were accepted as SHIV_{AD8-EO} RNA⁺ cells (Figure 3), SHIV_{AD8-EO} RNA⁺ virions (Figure 3), and bNAb⁺ cells (Figure 4) if they were present on at least 3 focal planes.

4. In our testing and optimization experiments to establish the RNAscope and bNAb imaging techniques, we scanned tissue sections that were 10-15 μm apart from each other in LN depth and obtained similar numbers of SHIV_{AD8-EO} RNA⁺ or bNAb⁺ cells.

Given the small sample size in the control (untreated) group and as indicated in the Figure legends, statistical analyses in Figure 3B and Supplemental Figure 4B was only performed between the WT bNAb-treated animals and DEL bNAb-treated animals at each timepoint or between timepoints for each bNAb-treated group, using the Mann-Whitney test. Figure 4 shows levels of LN bNAb⁺ cells in WT and DEL bNAb-treated monkeys and does not contain data from the control group.

Specific comment 4:

“Did the authors measure cell-associated viral DNA in LNs using PCR? It would be interesting to determine if there is any correlation with the results obtained from in situ hybridization analysis.”

Response:

We thank the reviewer for this suggestion and agree that it would be interesting to check if LN cell-associated viral DNA correlated with the RNAscope data we obtained. However, we did not have LN cells available for this analysis.

Specific comment 5:

“Considering the relevance of the RNA sequencing analysis in supporting the authors' conclusions regarding immune response modulation, it would be beneficial to assess the robustness of the findings by applying a more stringent fold-change cut-off (e.g., 1.5).”

Response:

We appreciate the comment of the reviewer regarding the relevance of the RNA sequencing analysis to our interpretation of immune modulation by bNAb administration. The RNA sequencing results presented in the manuscript (Figure 6 and Supplemental Figures 9 and 10) were obtained by Gene Set Enrichment Analysis (GSEA), which used as input the full list of genes ranked by their differential expression between monkey groups. Rank-based methods like GSEA should not be run on a list of genes filtered by an arbitrary cutoff. However, to demonstrate that our results can be reproduced with an alternative pathway enrichment analysis, we performed an overrepresentation analysis (ORA) using as input genes that were differentially regulated between monkey groups using the cutoff suggested by the reviewer (i.e. $|\log_2\text{FC}| > \log_2(1.5)$ & nominal p-value ≤ 0.05). As detailed below, the interpretation of the pathway enrichment analysis using ORA is the same as the one presented in the manuscript using GSEA.

To describe the GSEA results for monocytes and NK cells, we included the following sentences in the manuscript (page 20, lines 415-425):

“Gene set enrichment analysis (GSEA) revealed that LN monocytes and NK cells from treated monkeys significantly upregulated antiviral IFN- α signaling at week 8 post-challenge, but not earlier in infection, when compared to the pre-challenge timepoint (Figure 6, A and B). At week 2 post-challenge, monocytes from DEL bNAb-treated monkeys and NK cells from both WT and DEL bNAb-treated monkeys significantly upregulated TNF signaling via NF- κB (Figure 6, A and B and Supplemental Figure 9, A and B). However, such upregulation only persisted at week 8 in DEL bNAb-treated monkeys, whereas it was absent from

WT bNAb-treated and untreated monkeys at the same timepoint (Figure 6, A and B and Supplemental Figure 9, A and B). Also at week 8 post-challenge, IL-6/JAK/STAT3 signaling was upregulated exclusively in monocytes from DEL bNAb-treated monkeys (Figure 6A).”

Like GSEA, ORA revealed that, in monocytes from WT and DEL bNAb-treated monkeys, IFN- α signaling was significantly enriched among genes induced at week 8, but not at week 2 post-challenge, when compared to the pre-challenge timepoint (Figure R2). TNF signaling via NF- κ B was significantly enriched among genes induced in DEL bNAb-treated animals at weeks 2 and 8 post-challenge, but not in WT bNAb-treated monkeys at any timepoint (Figure R2). IL-6/JAK/STAT3 signaling was significantly enriched among genes induced in only DEL bNAb-treated animals at week 8 post-challenge (Figure R2).

Figure R2. Alternative to Figure 6A using ORA.

In NK cells and like GSEA, ORA revealed that IFN- α signaling was significantly enriched among genes induced in WT and DEL bNAb-treated monkeys at week 8, but not at week 2 post-challenge, when compared to the pre-challenge timepoint (Figure R3). TNF signaling via NF- κ B was significantly enriched among genes induced in WT and DEL bNAb-treated monkeys at week 2 post-challenge, but only remained significant at week 8 post-challenge in DEL bNAb-treated monkeys (Figure R3).

Figure R3. Alternative to Figure 6B using ORA.

To describe the GSEA results for T cells, we included the following sentences in the manuscript (page 21, lines 434-443):

“GSEA on the T cell transcriptomic dataset of untreated and DEL bNAb-treated monkeys revealed that antiviral IFN- α signatures were upregulated in all studied T cell types only when the virus was already detectable in plasma and LNs (i.e., at week 2 post-challenge in untreated monkeys and at week 8, but not week 2, post-challenge in DEL bNAb-treated monkeys) (Figure 6C). Interestingly, exclusively in DEL bNAb-treated monkeys at week 8 post-challenge, cell cycle-related genes were significantly upregulated in most non-naïve CD4⁺ and CD8⁺ T cell subsets (Figure 6C and Supplemental Figure 10A), and IL-2/STAT5 signaling and TNF signaling via NF- κ B were significantly downregulated in virtually all T cell subsets (Figure 6C and Supplemental Figure 10B).”

Like GSEA, ORA showed that in T cells, IFN- α signaling was significantly enriched among genes induced in DEL bNAb-treated monkeys at week 8 post-challenge and untreated monkeys at week 2 post-challenge (i.e. at timepoints of detectable virus), when compared to DEL bNAb-treated monkeys at week 2 post-challenge (when no virus was detected) (Figure R4). Cell cycle-related genes (E2F targets/G2/M checkpoint-related genes) were significantly enriched among genes induced in DEL bNAb-treated

monkeys at week 8 post-challenge only (Figure R4). TNF signaling via NF-κB and IL-2/STAT5 signaling were not significantly enriched among the genes that were differentially expressed (Figure R4).

Figure R4. Alternative to Figure 6C using ORA.

Specific comment 6:

“The authors concluded that the study highlights the usefulness of including bNABs with increased affinity to FcγRs to modulate the host immune response in future interventions for the treatment of HIV-1 infection. While I agree with the authors that further exploration of the effects and mechanisms of action of such bNABs should be encouraged, one might wonder about their added value considering that the overall viral burden was not affected and was even worse than when using WT bNABs. This aspect deserves further discussion.”

Response:

We thank the reviewer for this comment. One of the aims of our study was to test whether DEL bNABs given early after challenge could boost antiviral effects; indeed, they did not reduce viral burden and were not sufficient to mediate long-term viral control. We noted this in several parts of the Discussion, namely on page 24, lines 516-518; page 25, line 535-536, and page 26, lines 556-557. We agree with the reviewer that the lack of viral control in DEL bNAB-treated monkeys raises questions about the value of using them in experimental SHIV therapy settings or future HIV-1 therapy regimens, thus we have modified our concluding statement at the end of the Discussion (page 27, lines 585-586). Nonetheless, we also think that our findings on the effect of DEL bNAB infusions on T cells, monocytes, and NK cells are novel and relevant. The response profile of SIV Gag-specific CD8⁺ T cells and the proinflammatory signatures of LN monocytes and NK cells induced in DEL bNAB-treated monkeys but not in untreated monkeys should motivate further research on the cellular effects of Fc-modified bNABs and their mechanisms of action.

We have now added these discussion points in the revised version of the manuscript (page 26, lines 551-555).

Furthermore, as discussed on pages 26-27, lines 556-568, our study highlights the urgent need to gain a better understanding of the traffic and dynamics of infused bNAbs. Many studies have infused bNAbs in animal models and humans, but few have explored their distribution patterns in immunologically important sites such as the lymph nodes. By doing so in our study, we found that there was a temporal and spatial dissociation between virus and LNs that may have hindered formation of immune complexes and contributed to the lack of efficiency of the DEL bNAbs, but future studies are needed to explore this aspect.

Specific comment 7:

“Minor enhancements include specifying the type/origin of LNs analyzed and providing details on the number of replicates and experiments performed in Table 1.”

Response:

The following LNs were sampled at each timepoint: pre-challenge - left inguinal LNs, week 2 post-challenge – right inguinal LNs, week 8 post-challenge – left axillary LNs from WT and DEL bNAb-treated monkeys and left or right axillary LNs from untreated monkeys, and week 20 post-challenge – right axillary LNs. This information was included in the Methods (page 28, lines 602-606).

For the neutralization assays with results shown in Table 1, VRC07-523-LS, VRC07-523-LS/DEL, PGT121, and PGT121/DEL were tested against SHIV_{AD8-EO} in at least 5 experiments, with no more than 2-fold variability between them; representative IC₅₀ and IC₈₀ values are shown in Table 1. Dilution points for each bNAb were run in duplicate. Data for 3BNC117 and 10-1074 were obtained from Pegu et al., Cell Host & Microbe, 2019, PMID: 31513771. In the revised version of the manuscript, we included this information in two footnotes under Table 1 (page 61, lines 1323-1326).

Reviewer #2

General comments:

“Dias et al. have conducted a study in rhesus macaques examining how modifying broadly neutralizing antibodies (bNAbs) to better bind to Fc-gamma receptors with the S239D/I332E/A330L (DEL) mutation set affects early treatment of SHIV-AD8-EO infection with bNAbs and the endogenous immune response of the animals. This study builds on the scholarship surrounding bNAb treatment for SHIV infection in non-human primates (NHPs), specifically treatment during the early phase of infection as opposed to the chronic phase. Specifically, the main experiment of this manuscript was based upon the Nishimura et al. (Nature, 2017) study wherein NHPs were challenged with the same virus and given a combination of the wild-type bNAbs 3BNC117 and 10-1074 at days 3, 10, and 17 post-challenge during acute infection. In Dias et al., the macaques were challenged with SHIV-AD8-EO and given the antibodies VRC07-523-LS and PGT121 in either their wild-type state or with the DEL mutations. While many studies have also examined the effects of modifying the ability of bNAbs to bind to Fc receptors, this study appears to be the first to examine in vivo both modifications of the bNAbs and their effects during acute SHIV infection in NHPs.

Unlike in Nishimura et al. (Nature, 2017) study, where approximately half of the animals were able to immunologically control the virus, however, Dias et al. found that none of the macaques treated with the antibody combination had durable virologic control. Interestingly, the macaques given the bNAb combination with the DEL mutation had a higher proportion of SIV Gag-specific, CD69+ CD8 T cells producing IFN γ or IFN γ and MIP-1b. The macaques treated with the DEL mutation also had upregulated proinflammatory signatures in lymph node monocytes and increased plasma MIP-1b expression compared to the sham and wild-type bNAb-treated group. As such, this manuscript contains noteworthy data that is of significance to the study of bNAb treatment in SHIV (and HIV) infection, and I believe that the manuscript should be accepted with some revisions.”

Response:

We thank the reviewer for the positive assessment of our manuscript.

Major comment:

“I have one major concern regarding this study that I believe should be addressed. The addition of the DEL mutation to VRC07-523-LS resulted in a rather extreme difference in peak plasma concentration from 113 ug/mL with the wild-type to 15 ug/mL with the DEL mutant. While some of this difference could be explained by more efficient binding of the antibody to Fc receptors in the case of the DEL mutant, the authors also presented data showing that, at weeks 2 and 8 post-challenge, there was no significant difference in the number of cells with bound VRC07-523-LS wild-type or DEL in the lymph nodes. The PGT121 peak plasma concentrations were far more reasonable with a difference of 139 ug/mL versus 56 ug/mL in the wild-type versus DEL mutant, and the significant difference in the number of lymph node cells with bound PGT121 is much more convincing for the argument that the difference in plasma antibody concentration is likely owed to the enhanced Fc receptor binding. The nearly log difference in plasma concentration between the two versions of VRC07-523-LS just needs slightly more explanation. The human clinical trial detailed in Julg et al. (Nat Med, 2022), where viremic persons living with HIV were given triple bNAb therapy of PGT121, PGDM1400, and VRC07-523-LS, showed little to no development of resistance to VRC07-523-LS in the individuals’ viral populations despite relatively high plasma VRC07-523-LS concentrations while the viruses at the same time had become completely resistant to at least one of the other two antibodies when PGDM1400 and/or PGT121 plasma concentrations dipped below approximately 10 ug/mL. While I acknowledge that this is somewhat of a minor quibble and doesn’t have

much of an effect on the main conclusions of the paper, the difference in the plasma concentrations of the VRC07-523-LS variants is simply too large to not discuss it somewhat more.”

Response:

We agree with the reviewer that the difference in plasma concentration of VRC07-523-LS and its DEL mutant should be discussed and added a few sentences in the Discussion section to address this point (pages 25-26, lines 540-551). We concur with the reviewer that the higher affinity of VRC07-523-LS/DEL to FcγRs may not explain alone the differences in plasma levels between VRC07-523-LS/DEL and VRC07-523-LS. Of note, despite the substantially lower plasma levels of VRC07-523-LS/DEL when compared to its WT counterpart, we would like to point out that VRC07-523-LS/DEL was present between weeks 1 and 3 post-challenge at plasma levels that were, on average, 6- to 13- fold above the IC₈₀ and IC₅₀ titers for *in vitro* neutralization of the SHIV_{AD8-E0} challenge stock (Table 1), respectively. In addition, a single intravenous administration of VRC07-523-LS/DEL at 20 mg/kg was previously shown to decrease plasma viral loads in monkeys chronically infected with SHIV_{SF162P3} (Asokan et al., PNAS, 2020, PMID: 32690707). Also in that study, VRC07-523-LS/DEL plasma levels were lower than those of VRC07-523-LS throughout the first 5 days after bNAb infusion, but remained well above the IC₈₀ titers for *in vitro* SHIV_{SF162P3} neutralization (Asokan et al., PNAS, 2020, PMID: 32690707).

We would also like to acknowledge the clinical observation pointed out by the reviewer on the development of escape mutations to PGDM1400 and PGT121 as opposed to VRC07-523-LS in HIV-infected, viremic participants that were given the triple bNAb combination (Julg et al., Nat Med, 2022, PMID: 35551291). In our study, we did not find evidence for development of escape mutations to any of the infused bNAbs in the virus isolated at the time of peak viremia (Supplemental Table 4).

Minor comments

Specific comment 1:

“1. In Figure 6, Panel C, the image would be slightly more legible if the key for the comparisons was above the NES key and made larger or if the comparison was labelled on the actual heatmap where the color bands are at the bottom of the heatmap.”

Response:

We thank the reviewer for this comment and have updated Figure 6C; the key for the comparisons is now larger and positioned above the NES key.

Specific comment 2:

“2. In the legend of Supplemental Figure 1, maybe indicate that “DN” means double-negative with the rest of the abbreviations.”

Response:

We thank the reviewer for pointing this out and have now defined the “DN” abbreviation in the legend of Supplemental Figure 1.

Specific comment 3:

“3. In Supplemental Figure 7, Panel B, moving the antibody labels to below the divisions of the graphs instead of just using different symbols would make them easier to interpret and more legible.”

Response:

We thank the reviewer for this comment and have updated Supplemental Figure 7B; the antibody labels are now positioned under each graph.

Reviewer #3

General comments:

“The main message of the manuscript is the effect of two Fc-mutated bNAbs in the shaping of immune responses in the context of acute SHIV infection. However, the results do not formally support this conclusion.

The study presents a significant amount of data, yet a substantial portion of the six main figures focuses on aspects such as viral load, pharmacokinetics, ADA, virus and antibody localization. Regrettably, these data, while interesting, appear disconnected from the primary message of the paper, which aims to elucidate the modulation of immune responses by Fc-engineered antibodies.”

Response:

As stated in several parts of the manuscript (Abstract, page 4, lines 52-54; Introduction, page 6, lines 110-115; Discussion, page 22, lines 457-460), the aims of this study were to investigate the virological and immunological effects of administering to monkeys in the acute phase of SHIV_{AD8-EO} infection either WT bNAbs or bNAbs with the DEL mutation, which increases binding to FcγRs. We aimed to carry out this investigation not only in peripheral blood but also in lymph nodes given that these are relevant sites of HIV replication.

The data presented in the six main figures align with the study goals stated above. Specifically, studying the virological effects of early bNAb administration in peripheral blood and lymph nodes meant investigating if and how viral loads were affected by this therapy regimen, and we presented viral loads in plasma and lymph nodes in Figures 1 and 3, respectively. To understand the patterns of virus emergence in plasma, it was essential to measure the levels of infused bNAbs and ADA in circulation (Figure 2). Studying the immunological effects of early bNAb administration is a broad goal, and we focused on investigating if and how virus-specific T cell responses (Figure 5) and innate (monocytes and NK cells) and adaptive (T cells) cells were affected by this therapy regimen (Figure 6). For the immunological studies in lymph nodes (Figures 5 and 6), it was important to know whether the infused bNAbs reached these sites, thus we also measured bNAb levels in lymph nodes (Figure 4). For ease of read and flow, the manuscript was structured to first present the viremia and bNAb data in plasma (Figures 1 and 2, respectively), followed by the viremia and bNAb data in lymph nodes (Figures 3 and 4, respectively), and finally the immunological data in peripheral blood and lymph nodes (Figures 5 and 6).

It is also noteworthy that, within the aforementioned goals set for this study, there were two components that, to our knowledge, no other study has addressed so far: 1) the infusion of bNAbs with increased affinity to FcγRs in the acute phase of SHIV infection and 2) the detailed investigation of the presence, localization, and cellular effects of infused bNAbs in lymph nodes. The main figures of this manuscript include data on these two novel components.

The manuscript is rich in data from the virological and immunological points of view and given the novelty of the therapy setting that used Fc-mutated bNAbs early in SHIV infection, we think that all main figures are relevant, align with our study goals, and altogether lead to the conclusion that DEL bNAbs shape innate and adaptive cellular immunity.

Specific comment 1:

“The unexpected findings regarding the pharmacokinetics of antibodies are notable. Specifically, the observation that the LS mutant (VRC07-LS) does not exhibit prolonged serum concentrations compared to the unmutated Fc antibody (PGT121) warrants further investigation. Additionally, the striking influence of DEL mutation on the appearance of ADA is intriguing, yet the paper lacks an explanation for this phenomenon. Worthy of note, the appearance of ADA upon PGT121 and LS-mutated anti-HIV-1 bNAbs has already been reported (Hsu et al., 2021; PLoS Pathog 17(2): e1009339. <https://doi.org/10.1371/journal.ppat.1009339>).”

Response:

Introduction of the LS mutation in a bNAb prolongs its serum concentration and extends its half-life *in vivo* when compared to its parental unmodified bNAb that does not contain the LS mutation. This has been shown in NHPs and humans for different pairs of bNAbs, such as VRC01 vs. VRC01LS, VRC07 vs. VRC07-LS, N6 vs. N6-LS, 3BNC117 vs. 3BNC117-LS, PGDM1400 vs. PGDM1400-LS, and 10-1074 vs. 10-1074-LS (Ko et al., Nature, 2014, PMID: 25119033; Gaudinski et al., PLOS Med, 2018, PMID: 29364886; Rudicell et al., J Virol, 2014, PMID: 25142607; Julg et al., J Virol, 2017, PMID: 28539448; Mayer et al., Pharmaceutics, 2024, PMID: 38794258).

Different bNAbs have different pharmacokinetic (PK) profiles (Shingai et al., JEM, 2014, PMID: 25155019; Julg et al., Sci Transl Med, 2017, PMID: 28931655; Mayer et al., Pharmaceutics, 2024, PMID: 38794258), thus it is not fair to assess the effect of the LS mutation in bNAb half-life extension by comparing the PK profile of VRC07-523-LS with that of a different bNAb (in this case PGT121) without the LS mutation. Moreover, one of the reasons for having selected VRC07-523-LS and PGT121 for this study was precisely because they have similar PK profiles upon intravenous infusion in NHPs, as opposed to VRC07-523 and PGT121 (Hessell et al., Nat Med, 2016, PMID: 26998834). The observation that VRC07-523-LS and PGT121 have similar PK profiles is based on previous studies (Barouch et al., Nature, 2013, PMID: 24172905; Julg et al., J Virol., 2017, PMID: 28539448; Rudicell et al., J. Virol., 2014, PMID: 25142607; Julg et al., Sci Transl Med, 2017, PMID: 28931655; Sobieszczyk et al., Lancet HIV, 2023, PMID: 37802566), was stated in the Introduction of our manuscript (page 7, lines 120-121), and confirmed in our study (Figure 2, A and B).

Although we agree with the reviewer that the development of stronger ADA responses against DEL bNAbs when compared to wild-type (WT) bNAbs is intriguing, the investigation of the mechanism(s) through which the DEL mutation triggered ADA responses is beyond the scope of this study. Nonetheless, we have pointed out this intriguing observation in the Discussion (page 23, lines 494-495) and added the study indicated by the reviewer (Hsu et al., PLOS Pathog, 2021, PMID: 33600506) as a support for the development of ADA in NHPs following infusion of human bNAbs (page 24, line 496).

Specific comment 2:

“The data pertaining to the shaping of immune response by Fc-mutated bNAbs are insufficient and fail to provide convincing evidence. For instance, the frequency of MIP-1 β expressing cells, which is lower than 0.1%, is practically negligible. To derive more robust conclusions, appropriate controls, such as PBMC activation by an irrelevant peptide, are imperative. Furthermore, the concentrations of MIP-1 β detected in the plasma of DEL-treated animals do not support an antibody-mediated effect on its production by CD8 cells. Notably, MIP-1 β can be secreted by different immune cells and it is mostly detected at weeks 2-3 post-challenge, while MIP-1 β -secreting CD8 T cells are identified at 20 weeks post-challenge. Moreover, the frequency of IFN- γ expressing cells in DEL-treated individuals appears to be both notably low and quite variable, with only a minority of animals exhibiting frequencies higher than those in other groups. It would have been enlightening to establish a relationship between these "responding" animals

and their viral loads, as the frequency of SIV Gag-specific CD8 cells appears to correlate more closely with viral rebound than with the antibody treatment itself.”

Response:

We agree with the reviewer that the frequency of SIV Gag-specific CD8⁺ T cells expressing MIP-1 β is very low and likely negligible. Thus, we have further toned down our statement related to these data in the Results section (page 18, lines 374-377). We have also removed the association of MIP-1 β production to viral suppression and our T cell RNA sequencing data from the Discussion, which also helped us address this reviewer’s specific comment 4. In addition, we removed the relevance attributed to these findings by taking them out of the Abstract and the first and third paragraphs of the Discussion.

Regarding the controls used in the functional assays, each sample (PBMC or LN cell sample) at each timepoint was incubated with either SIV Gag peptide pool (SIV Gag-stimulated sample) or DMSO, which was the SIV Gag peptide pool vehicle (negative control). Incubating cells with DMSO is an appropriate control because it allows residual, background/non-specific response levels to be subtracted from the response levels of the corresponding SIV Gag-stimulated samples, resulting in response levels that are specific to the SIV Gag peptide pool. This is the standard for T cell immunology throughout the literature. All data presented in Figure 5 and Supplemental Figure 6, B and C, was background subtracted and these experimental details are included in the Methods (subsection “*In vitro* cell assays”, pages 37-38).

Regarding MIP-1 β production, we did report that DEL bNAb-treated monkeys had higher levels of MIP-1 β in plasma during and shortly after the bNAb infusions but did not intend to infer that MIP-1 β was produced by CD8⁺ T cells at that time. This cytokine was measured in plasma samples by Luminex and we agree with the reviewer that it could have been produced by many different cell types. In the revised version of the manuscript, we have clarified our statement in the Results section (page 18, lines 377-378).

Regarding the reviewer’s comment on the frequency of IFN γ -expressing cells in DEL bNAb-treated monkeys (Figure 5C), we found no significant correlation between the frequency of circulating SIV Gag-specific CD69⁺IFN γ ⁺MIP-1 β ⁻TNF⁻CD107a⁻ CD8⁺ T cells and plasma viral loads at week 20 post-challenge (Spearman’s $r = 0.4286$, $p = 0.35$) (Figure R5). As noted in the Discussion (page 24), similar plasma viral loads among monkey groups at that timepoint rule out an effect of viremia alone in the different response pattern detected in DEL bNAb-treated monkeys when compared to untreated and WT bNAb-treated monkeys.

Figure R5. Correlation between the frequency of SIV Gag-specific CD69⁺IFN γ ⁺MIP-1 β ⁻TNF⁻CD107a⁻ CD8⁺ T cells in PBMCs and plasma viral load in DEL bNAb-treated monkeys at week 20 post-challenge. The Spearman’s correlation test was used.

Specific comment 3:

“The evidence supporting the effect of DEL treatment in shaping monocyte and NK cell properties is limited. Validating the upregulated pathways at the protein level, such as IL-6 and TNF production by monocytes, would strengthen the conclusions. In addition, the observed enhanced inflammatory response in DEL-treated animals at week 8 post-challenge may be attributed to the high viral load specifically observed in this group of animals at this timepoint, rather than the antibody treatment. Notably, monocytes secrete strong quantities of IL-6 and TNF upon stimulation by HIV-1 and SIV. Additionally, the speculation regarding DEL-antibodies' retention in LN due to FcγR-expressing cell binding lacks in vivo validation.”

Response:

We agree with the reviewer that measuring IL-6 and TNF production by monocytes would strengthen our conclusions. However, considering that monocyte viability is severely compromised by freeze-thaw cycles and we used a lot of samples for the experiments presented in the manuscript, we do not have the samples required to successfully conduct such analysis.

The proinflammatory signatures detected in DEL bNAb-treated monkeys at week 8 post-challenge in LN monocytes and NK cells (i.e. upregulation of TNF signaling via NF-κB in both monocytes and NK cells and upregulation of IL-6/JAK/STAT3 signaling in monocytes) cannot be solely attributed to high viral loads in these monkeys. Although WT bNAb-treated monkeys did not show virus until, on average, week 10 post-challenge, untreated monkeys had high viral loads at week 8 post-challenge (Figure 1); yet, such pro-inflammatory responses were not detected in the absence of bNAb treatment. This implicates DEL bNAb treatment as a driving factor of these response patterns.

In the revised version of the Discussion, we have removed the sentence where we had speculated that DEL antibodies could have been retained in the extrafollicular area through binding to FcγR-expressing cells residing there.

Specific comment 4:

“The transcriptomic analysis of T cells lacks clarity. The choice of group comparisons is ambiguous, and the purported downregulation of TNF and its association with reduced viral replication contradicts data on LN viral burden at week 8 post-challenge.”

Response:

In light of our RNA sequencing results that showed that at week 8 post-challenge, monocytes from DEL bNAb-treated monkeys upregulated proinflammatory signatures (Figure 6A), we aimed to investigate the effect of DEL bNAb infusions on LN T cells at the same timepoint. To assess the effect of DEL bNAbs on LN T cells at week 8 post-challenge without having viremia as a confounding factor, we compared the transcriptomic programs of T cells from DEL bNAb-treated monkeys at week 8 post-challenge to those from untreated monkeys at week 2 post-challenge since their viral loads were comparable. Samples from DEL bNAb-treated monkeys at week 2 post-challenge were used to assess the effect of DEL bNAbs longitudinally bearing in mind that monkeys were still aviremic at that timepoint. In the revised version of the manuscript, we included this justification in the Results section (page 21, lines 429-434).

We agree with the reviewer that associating the downregulation of TNF signaling via NF-κB on LN T cells to reduced viral replication does not align with our LN viral load data and we thank the reviewer for pointing this out. In the Discussion of the revised manuscript, we have removed the sentences where we had made that association.

Specific comment 5:

“The discussion is based on speculation, proposing hypothetical modulation of immune responses through bNAbs treatment in an Fc-dependent manner, without direct experimental support.”

Response:

We think the Discussion section is the right place to put results into context and make connections to previous studies; this is also the section where the authors usually have some freedom to raise hypotheses based on the results obtained that can help plan future studies. Nonetheless, in the revised version of the manuscript, we have removed speculative sentences from the Discussion based on this reviewer’s comments 2, 3, and 4. We have also enriched the Discussion with important points raised by all reviewers.

Specific comment 6:

“In summary, despite the extensive dataset, the results do not sufficiently support the conclusion that Fc-engineered antibodies modulate adaptive and innate immune responses in SHIV-infected animals. Moreover, the lack of efficient viral control by the bNAbs used in this study raises doubts regarding their potential to modulate antiviral host immune responses effectively.”

Response:

Through the results presented in this manuscript as well as the improvements made and clarifications provided in response to the reviewers’ questions, we reported response patterns and transcriptomic signatures on T cells, monocytes, and NK cells that are attributed to the infusion of DEL bNAbs. The results presented and conclusions drawn are based on a well-controlled study that was carefully designed and experiments that were rigorously planned and executed.

Regarding the lack of efficient viral control observed in our study and the doubts that arise in terms of bNAbs antiviral efficacy, we thank the reviewer for pointing this out. As we have addressed a similar concern in response to Reviewer 1, specific comment 6, the responses to that comment and this one are very similar.

One of the aims of our study was to test whether DEL bNAbs given early after challenge could boost antiviral effects; indeed, they did not reduce viral burden and were not sufficient to mediate long-term viral control. We noted this in several parts of the Discussion, namely on page 24, lines 516-518; page 25, line 535-536, and page 26, lines 556-557. We agree with the reviewer that the lack of viral control in DEL bNAbs-treated monkeys raises questions about their antiviral efficacy and the value of using them in experimental SHIV therapy settings or future HIV-1 therapy regimens, thus we have modified our concluding statement at the end of the Discussion (page 27, lines 585-586). Nonetheless, we also think that our findings on the effect of DEL bNAbs infusions on T cells, monocytes, and NK cells are novel and relevant. The response profile of SIV Gag-specific CD8⁺ T cells and the proinflammatory signatures of LN monocytes and NK cells induced in DEL bNAbs-treated monkeys but not in untreated monkeys should motivate further research on the cellular effects of Fc-modified bNAbs and their mechanisms of action. We have now added these discussion points in the revised version of the manuscript (page 26, lines 551-555).

Furthermore, as discussed on pages 26-27, lines 556-568, our study highlights the urgent need to gain a better understanding of the traffic and dynamics of infused bNAbs. Many studies have infused bNAbs in animal models and humans, but few have explored their distribution patterns in immunologically important sites such as the lymph nodes. By doing so in our study, we found that there was a temporal and spatial dissociation between virus and LNs that may have hindered formation of immune complexes and contributed to the lack of efficiency of the DEL bNAbs, but future studies are needed to explore this aspect.

Reviewer #4

General comments:

“I co-reviewed this manuscript with one of the reviewers who provided the listed reports. This is part of the Nature Communications initiative to facilitate training in peer review and to provide appropriate recognition for Early Career Researchers who co-review manuscripts”

Response:

We thank the reviewer for co-reviewing our manuscript.

Reviewers' Comments:

Reviewer #1:

Remarks to the Author:

I thank the authors for their thorough and diligent revisions to the manuscript. I appreciate their effort in addressing all the points I raised in my initial review. The detailed responses and the subsequent revisions have significantly improved the clarity and robustness of the manuscript. It is unfortunate that LN samples are not available at the week 10 to 12 timepoints to generate the requested data as per comment 1. I understand the constraints and limitations.

The additional data and analyses provided in other areas have been very helpful and contribute substantially to the overall quality of the study.

I believe the manuscript is now much stronger and will be a valuable contribution to the field.

Reviewer #2:

Remarks to the Author:

Dias et al have conducted a study in rhesus macaques examining how modifying broadly neutralizing antibodies (bNAbs) to better bind to Fc-gamma receptors with the S239D/I332E/A330L (DEL) mutation set affects early treatment of SHIV-AD8-EO infection with bNAbs and the endogenous immune response of the animals. As I wrote in my original review, I find the manuscript to contain noteworthy data of significance to the study of bNAb treatment in SHIV and thus HIV infection. I am completely happy with the changes made to the paper after review and fully support the acceptance of the manuscript.

Sincerely,

Victoria Walker-Sperling

Reviewer #3:

Remarks to the Author:

The authors have addressed the reviewers' comments by removing several conclusions from the article that were not formally demonstrated and by moderating some of the manuscript's statements.

However, the main issue remains that the bNAbs used do not provide efficient viral control. While some differences in the innate and adaptive immune responses were observed in DEL bNAb-treated monkeys, the "shaping of the immune response" is insufficient to effectively modulate antiviral host immune responses. This issue is now further discussed.

Reviewer #4:

Remarks to the Author:

The authors made modifications to improve the accuracy of their conclusions.

In this sense, a more precise title would be appreciated. Accordingly, the authors could consider a sentence summarizing their study, which focuses on the presence, localization, and cellular effects of anti-HIV-1 bNAbs with increased affinity to FcγRs administered during the acute phase of SHIV infection. However, despite the presence of effector functions, these were not sufficient to induce long-term viral control in DEL bNAb-treated monkeys.

Reviewer #1

“I thank the authors for their thorough and diligent revisions to the manuscript. I appreciate their effort in addressing all the points I raised in my initial review. The detailed responses and the subsequent revisions have significantly improved the clarity and robustness of the manuscript.

It is unfortunate that LN samples are not available at the week 10 to 12 timepoints to generate the requested data as per comment 1. I understand the constraints and limitations.

The additional data and analyses provided in other areas have been very helpful and contribute substantially to the overall quality of the study.

I believe the manuscript is now much stronger and will be a valuable contribution to the field.”

Response:

We thank the reviewer for the positive assessment of our manuscript.

Reviewer #2

“Dias et al have conducted a study in rhesus macaques examining how modifying broadly neutralizing antibodies (bNAbs) to better bind to Fc-gamma receptors with the S239D/I332E/A330L (DEL) mutation set affects early treatment of SHIV-AD8-EO infection with bNAbs and the endogenous immune response of the animals. As I wrote in my original review, I find the manuscript to contain noteworthy data of significance to the study of bNAb treatment in SHIV and thus HIV infection. I am completely happy with the changes made to the paper after review and fully support the acceptance of the manuscript.”

Response:

We thank the reviewer for the positive assessment of our manuscript.

Reviewer #3

“The authors have addressed the reviewers' comments by removing several conclusions from the article that were not formally demonstrated and by moderating some of the manuscript's statements.

However, the main issue remains that the bNAbs used do not provide efficient viral control. While some differences in the innate and adaptive immune responses were observed in DEL bNAb-treated monkeys, the "shaping of the immune response" is insufficient to effectively modulate antiviral host immune responses. This issue is now further discussed.”

Response:

We thank the reviewer for the positive assessment of our manuscript.

Reviewer #4

“The authors made modifications to improve the accuracy of their conclusions.

In this sense, a more precise title would be appreciated. Accordingly, the authors could consider a sentence summarizing their study, which focuses on the presence, localization, and cellular effects of anti-HIV-1 bNAbs with increased affinity to FcγRs administered during the acute phase of SHIV infection. However, despite the presence of effector functions, these were not sufficient to induce long-term viral control in DEL bNAb-treated monkeys.”

Response:

We thank the reviewer for the positive assessment of our manuscript. We have updated the title to the one suggested by the journal; the new title is in agreement with the reviewer's comment and fits the journal guidelines.